# Ewing Sarcoma—Diagnosis, Treatment, Clinical Challenges and Future Perspectives

**DOI:** 10.3390/jcm10081685

**Published:** 2021-04-14

**Authors:** Stefan K. Zöllner, James F. Amatruda, Sebastian Bauer, Stéphane Collaud, Enrique de Álava, Steven G. DuBois, Jendrik Hardes, Wolfgang Hartmann, Heinrich Kovar, Markus Metzler, David S. Shulman, Arne Streitbürger, Beate Timmermann, Jeffrey A. Toretsky, Yasmin Uhlenbruch, Volker Vieth, Thomas G. P. Grünewald, Uta Dirksen

**Affiliations:** 1Pediatrics III, University Hospital Essen, 45147 Essen, Germany; Uta.Dirksen@uk-essen.de; 2West German Cancer Center (WTZ), University Hospital Essen, 45147 Essen, Germany; sebastian.bauer@uk-essen.de (S.B.); stephane.collaud@rlk.uk-essen.de (S.C.); jendrik.hardes@uk-essen.de (J.H.); Arne.Streitbuerger@uk-essen.de (A.S.); beate.timmermann@uk-essen.de (B.T.); 3German Cancer Consortium (DKTK), Essen/Düsseldorf, University Hospital Essen, 45147 Essen, Germany; 4Cancer and Blood Disease Institute, Children’s Hospital Los Angeles, Keck School of Medicine, University of Southern California, Los Angeles, CA 90027, USA; jamatruda@chla.usc.edu; 5Department of Medical Oncology, Sarcoma Center, University Hospital Essen, 45147 Essen, Germany; 6Department of Thoracic Surgery, Ruhrlandklinik, University of Essen-Duisburg, 45239 Essen, Germany; 7Institute of Biomedicine of Sevilla (IbiS), Virgen del Rocio University Hospital, CSIC, University of Sevilla, CIBERONC, 41013 Seville, Spain; enrique.alava.sspa@juntadeandalucia.es; 8Department of Normal and Pathological Cytology and Histology, School of Medicine, University of Seville, 41009 Seville, Spain; 9Dana-Farber/Boston Children’s Cancer and Blood Disorders Center, Harvard Medical School, Boston, MA 02215, USA; Steven_DuBois@dfci.harvard.edu (S.G.D.); David_Shulman@dfci.harvard.edu (D.S.S.); 10Department of Musculoskeletal Oncology, Sarcoma Center, 45147 Essen, Germany; 11Division of Translational Pathology, Gerhard-Domagk Institute of Pathology, University Hospital Münster, 48149 Münster, Germany; wolfgang.hartmann@ukmuenster.de; 12West German Cancer Center (WTZ), Network Partner Site, University Hospital Münster, 48149 Münster, Germany; 13St. Anna Children’s Cancer Research Institute and Medical University Vienna, 1090 Vienna, Austria; heinrich.kovar@ccri.at; 14Department of Pediatrics and Adolescent Medicine, University Hospital Erlangen, 91054 Erlangen, Germany; Markus.Metzler@uk-erlangen.de; 15Department of Particle Therapy, University Hospital Essen, West German Proton Therapy Centre, 45147 Essen, Germany; 16Departments of Oncology and Pediatrics, Georgetown University, Washington, DC 20057, USA; jat42@georgetown.edu; 17St. Josefs Hospital Bochum, University Hospital, 44791 Bochum, Germany; yasmin.uhlenbruch@klinikum-bochum.de; 18Department of Radiology, Klinikum Ibbenbüren, 49477 Ibbenbühren, Germany; v.vieth@klinikum-ibbenbueren.de; 19Division of Translational Pediatric Sarcoma Research, Hopp-Children’s Cancer Center Heidelberg (KiTZ), 69120 Heidelberg, Germany; t.gruenewald@kitz-heidelberg.de; 20Division of Translational Pediatric Sarcoma Research, German Cancer Research Center (DKFZ), 69120 Heidelberg, Germany; 21Institute of Pathology, University Hospital Heidelberg, 69120 Heidelberg, Germany; 22German Cancer Consortium (DKTK), Core Center, 69120 Heidelberg, Germany

**Keywords:** ewing sarcoma, small round cell sarcoma, limb salvage, metastasis, EWSR1-FLI1, chromosomal translocation, fusion protein, transcription, splicing

## Abstract

Ewing sarcoma, a highly aggressive bone and soft-tissue cancer, is considered a prime example of the paradigms of a translocation-positive sarcoma: a genetically rather simple disease with a specific and neomorphic-potential therapeutic target, whose oncogenic role was irrefutably defined decades ago. This is a disease that by definition has micrometastatic disease at diagnosis and a dismal prognosis for patients with macrometastatic or recurrent disease. International collaborations have defined the current standard of care in prospective studies, delivering multiple cycles of systemic therapy combined with local treatment; both are associated with significant morbidity that may result in strong psychological and physical burden for survivors. Nevertheless, the combination of non-directed chemotherapeutics and ever-evolving local modalities nowadays achieve a realistic chance of cure for the majority of patients with Ewing sarcoma. In this review, we focus on the current standard of diagnosis and treatment while attempting to answer some of the most pressing questions in clinical practice. In addition, this review provides scientific answers to clinical phenomena and occasionally defines the resulting translational studies needed to overcome the hurdle of treatment-associated morbidities and, most importantly, non-survival.

## 1. Introduction

Ewing sarcoma (EwS) represents a rare, highly malignant cancer, with most patients harboring *a priori* micrometastases [1,2], since, without systemic therapy, over 90% of patients die from disseminated disease [3]. It is most commonly diagnosed in the second decade of life; however, patients have presented as early as newborn and as late as into the eighth decade, with tumors in almost every bodily location.

Current EwS therapy emphasizes a multimodal approach, which, as a result of collaborative trials, has led to improved overall survival (OS) for localized disease [4,5,6,7]. Despite multimodal treatment, survival in metastatic disease occurring in 20–25% of patients, predominantly in the lungs (70–80%) and bone/bone marrow (40–45%), is still associated with a dismal prognosis [8,9]. In addition, recurrent disease is observed in 30–40% of patients with primary non-metastatic disease, increasing to 60–80% for EwS patients with metastatic disease at diagnosis. Relapse is mostly systemic (71–73%), followed by combined (12–18%) and local (11–15%) relapse, leading to five-year post-relapse survival rates of 15–25%, with local recurrence faring better than systemic [10,11,12]. Systemic tumor control still poses the main therapeutic challenge.

To achieve significant improvement to overcome plateaued survival rates, especially for high-risk patients, innovative clinical strategies and novel therapeutic concepts are required. EwS provides a tumor-specific molecular target which is indispensable for tumor development. Characteristically, EwS carry a balanced translocation. In 85–95% of all EwS patients, this rearrangement fuses the Ewing sarcoma breakpoint region 1 gene (*EWSR1*) on chromosome 22 to the friend of leukemia virus integration site 1 gene (*FLI1*) on chromosome 11 t(11;22)(q24;q12) [13]. The resulting EWSR1-FLI1 fusion product functions as an oncoprotein that is both necessary and presumably sufficient for tumorigenesis [14,15]. Consequently, inactivation of EWSR1-FLI1 function is desirable for effective therapy, although it is clinically not mandatory, as shown by effectiveness of non-targeted chemotherapy in a substantial proportion of patients with localized tumors. 

Still, many aspects of the disease require further study, e.g., cryptic cell of origin, phenomenon of oncogene addiction as well as oncogene plasticity, distinct molecular activities and clinical relevance of fusion proteins in EwS, CIC-rearranged sarcoma, sarcoma with BCOR genetic alterations, and round cell sarcoma with EWSR1-non-ETS fusions (all together formerly known as “Ewing-like sarcoma”. This term refers to a morphological similarity, but falsely suggests both a similar genetic background and a clinical similarity, from hereafter termed as “related entities”) [16]. Although consensus between national and international guidelines for standard practice of patients with EwS would be desirable, the practical approach reveals differences in clinical care, especially in areas without clear evidence [17]. Therefore, the following review provides an overview of both the current standards and remaining questions in clinical practice of EwS. Clinical information is supplemented by scientific summaries to address EwS-specific clinical phenomena. Each section additionally tries to define the next steps in translational research to improve the standard of care for patients with EwS.

## 2. Diagnosis

### 2.1. Imaging (by V. Vieth)

#### 2.1.1. Diagnostic Workup—The Timeless Value of Plain Radiographs for Deciphering Bone Lesions 

The early diagnosis of EwS remains challenging. Despite similar symptoms, pseudotumoral and benign bone lesions occur more frequently [18,19].

The initial staging, the biopsy, both the local and systemic therapy, as well as the follow-up care are all based on the findings of the imaging. In consequence, choosing the appropriate imaging modality for patients with EwS is decisive for both diagnostic and therapeutic assessment, while delineating the treatment strategy.

Nowadays, the primary diagnostic work-up of bone pain, especially in children, requires magnetic resonance imaging (MRI) which exceeds a high negative predictive value for malignant bone tumors [20]. If the MRI shows inconclusive findings, a projection radiography or, in the case of locations that cannot be displayed without overlapping, a computed tomography (CT) must be carried out. Decades ago, Lodwick formulated the still valid meaning of projection radiography in the diagnosis of bone tumors: “… most of us, perhaps without recognizing a logical basis for such a decision, assign a certain growth rate or degree of malignancy … to a tumor based on its radiographic image” [21]. In projection radiography/CT, signs that are consistent with a suspected malignant bone tumor such as EwS include permeative osteolysis (stage III, classified according to Lodwick-Madewell [22], periosteal reactions with interrupted compacta (onion skin phenomenon, spiculae, Codman triangle) [23,24], and mineralization of matrix. Signs of malignancy on MRI include solid displacement of the bone marrow and the extraosseous tumor extension and joint infiltration. The MRI provides additional information regarding differential diagnoses. Choosing the right sequences is crucial: while the classic T1 and T2 contrast is indispensable, proton-weighted and gradient echo sequences do not help in tissue characterization [25]. In MRI, EwS presents as a solid tumor mass in bone with low signal intensity in T1 and high signal intensity in T2. There is often a sharp transition zone in the bone portion of the tumor. The tumor shows peritumoral edema and gadolinium enhancement. A soft-tissue mass is often present. MRI does not show specific signs that can include or exclude EwS compared with, for example, osteomyelitis [26].

EwS-related entities may differ radiologically from classical EwS, e.g., small round cell sarcomas (SRCS) with *CIC-DUX4* fusion often present as necrotic and hypermetabolic soft-tissue masses while SRCS with *BCOR-CCNB3* translocations are vascular bone lesions with necrosis at imaging (please see section “Round cell sarcoma with non-ETS-fusions and CIC/BCOR-rearranged sarcoma”) [27].

#### 2.1.2. Local Tumor Assessment and Staging—“Trust in T1”

Imaging guidelines for patients with EwS have been proposed [28]. The MRI is the method of choice for visualizing the local extent of the tumor. The native T1 sequence is best suited for determining the resection height [29]—“trust in T1”. The protocol must be supplemented by further sequences, i.e., T2 TSE and the T1 TSE sequence with contrast media and fat saturation, to address extraosseous tumor infiltration of adjacent vascular/nerve bundles or joint compartments as these findings impact the extent and technique of local therapy (Figure 1) [30,31].

To rule out skip metastases in the same bone or, rarely, in the adjacent bone [32], the entire bone compartment must additionally be imaged using the body coil and a coronal T1 and coronal STIR sequence. ^18^F-FDG-PET/CT with a diagnostic chest CT, and either ^18^F-FDG-PET/MRI or whole-body MRI, each combined with a thorax CT, are reliable diagnostics in staging of EwS patients. For the diagnosis of bone metastases, both ^18^F-FDG-PET/CT and ^18^F-FDG-PET/MRI are comparable and significantly more accurate than CT and bone scintigraphy. Combined administration of ^18^F-NaF/^18^F-FDG-PET/CT may further improve skeletal disease detection [33,34]. Whole-body MRI shows better results in the detection of bone metastases/multifocality compared to ^18^F-FDG-PET/CT in some studies with sensitivity rates of 94% compared to 78% and specificity rates of 76% compared to 80%, respectively [35]. The combination of whole-body MRI with chest CT will likely continue to gain acceptance and current PET-MRI hybrid systems might improve its specificity. Nevertheless, complementary chest CT will remain indispensable [36]. 

Complete imaging for both locoregional expansion (T-staging) and distant metastasis (M-staging) before the start of chemotherapy is crucial (Figure 1). Inconclusive findings must be clarified by further imaging or biopsy. Effects from chemotherapy such as bone marrow conversion and tumor response often make it impossible to assess findings after the onset of neoadjuvant chemotherapy.

#### 2.1.3. Therapeutic Assessment and Follow-Up

The radiological response to chemotherapy is important when considering local therapy options [17]. Follow-up care as part of trial protocols creates a survival advantage [37]. Both timing and modality of imaging are based on trial recommendations. The primary tumor is examined by MRI, the lungs by CT, and the entire body by ^18^F-FDG-PET/CT/MRI. Suspicious findings can guide therapeutic decision making (Figure 1) [38].

### 2.2. Biopsy (by E. de Álava, W. Hartmann, and V. Vieth)

#### 2.2.1. How to Biopsy in EwS?

The correct diagnosis of EwS remains crucial and requires an interdisciplinary approach (Figure 2). Following clinical suspicion and radiologically added confirmation, a variety of options are available for retrieving the necessary biological material to achieve a histological diagnosis of a suspected bone tumor [39,40]. The MRI provides the crucial information for biopsy planning by distinguishing solid tumor tissue, cysts, necroses, hemorrhages and extraosseous tumor components [41]. Different tumor parts should be biopsied representatively with knowledge of the imaging.

To evaluate histologic subtype, and add suitable molecular genetic analyses, which in turn guide the decision making for multimodality therapy, the workup of suspected sarcoma requires more material than can be obtained from a fine-needle aspiration [42]. Thus, either surgical incisional biopsy, i.e., open biopsy, or percutaneous core needle biopsy (CNB) with a CT/MRI scan is required for proper sarcoma diagnosis and to adequately define therapeutic strategies. Taking a sample from the extraosseous soft-tissue tumor is usually sufficient; removal of bone tissue is only necessary if the tumor is located in the bone. 

The biopsy method of choice in EwS remains controversial, since randomized controlled trials to compare CNB with the open biopsy procedure have not been conducted yet. Open or CNB biopsy is recommended if EwS is suspected. Moreover, suspected solitary bone metastases as well as metastases in lymph nodes should be biopsied at presentation if possible [17].

The accuracy of open biopsies is close to 100% in some reported publications [43,44,45]. CNB-reported biopsy success rates in sarcoma patients vary from 50% to 98% [46,47,48,49,50,51]. The success rate for needle biopsy may be inferior compared to open biopsy, specifically in EwS patients [51]. However, in experienced centers, the rate of sampling errors may be as low as for open biopsies. Importantly, EwS can imitate osteomyelitis both clinically (fever, increased infection values, isolated bone pain) and radiologically [51,52,53]. The possibility of a sampling error of the biopsy material with merely reactively altered tissue must always be considered and, if in doubt, renewed sampling must be considered. Irrespective of biopsy method, it is essential to avoid hematomas and contamination of neurovascular structures or joints, since all tissue that is considered to be contaminated must be resected afterwards if EwS is diagnosed.

In any case, biopsy procedures for suspected EwS should be performed at a specialist sarcoma referral center in consultation with the tumor orthopedic team that will carry out definitive tumor resection. Larger resections and amputations due to inappropriate needle biopsy technique, where limb salvage would otherwise have been possible, have been reported [54]. The biopsy site can be marked with a skin tattoo, which will allow its identification at time of surgery following neoadjuvant chemotherapy [55].

#### 2.2.2. The Risk of Tumor Seeding along the Access Path of Biopsy

Historically, the open biopsy technique has been associated with a significantly increased risk of tumor seeding along the biopsy tract when the scar was not removed en-bloc during surgical resection of the tumor [56]. However, even an open biopsy can be done by using a short incision (2–4 cm; in the area of the extremities, access must always be set lengthways), a small opening of the bone, a mandatory wound drainage in order to prevent hematoma and an intracutaneous suture technique. Circulating EwS cells have been demonstrated in blood during uncontaminated tumor removal, but no relationship to survival has been established [57,58]. CNB represents a safe, minimal invasive and cost-effective technique (shorter hospitalization) with presumed lower complication rates (infection, hematoma, fracture, reduction in therapy-free interval) [59]. The perceived potential for periinterventional tumor seeding along the CNB tract has not been fully elucidated yet [43,60]. In consequence, resection of the CNB tract is recommended by several authors [61,62], without sufficient evidence to support an increased risk of either tumor seeding along the CNB tract or local recurrence when the CNB tract is not resected [63,64,65]. Still, if bleeding occurs at CNB, it must not be tolerated. Until reliable data have ruled out an increased risk for tumor seeding following CNB in EwS, the biopsy needle track should be placed to attain the highest possible yield while minimizing contamination of normal tissues, and so that it can be incorporated into the final surgical excision [50].

#### 2.2.3. Biopsy—The Holy Tissue Grail

The recommendation for open biopsy also reflects the need for tissue on which to conduct research. Only open biopsy carries the advantage to provide sufficient material for both histological diagnosis and translational research on tumor tissue prior systemic treatment. Therapy-naïve tumor specimens appear crucial for preclinical drug testing and molecular studies to ultimately improve EwS patient survival. In case of first EwS relapse, re-biopsy is often recommended to confirm definitive diagnosis of relapse as well as to provide tissue for both genetic testing for targetable mutations and research questions within the frame of collaborative translational projects. One drawback is the possible elimination of measurable or evaluable disease that may be required for clinical trial enrollment or assessment of response to therapy. While diagnostic confirmation by re-biopsy appears crucial, it remains elusive if EwS patients ultimately benefit from target analysis to identify molecular actionable variants in relapsed situation [66].

### 2.3. Pathological Diagnosis (by E. de Álava, T. G. Grünewald, and W. Hartmann)

#### 2.3.1. How to Diagnose EwS?

The definitive diagnosis of EwS should be made (or reviewed) at a sarcoma reference center by biopsy, providing sufficient material for conventional histology, immunohistochemistry, molecular pathology and biobanking (please see section “Biopsy—the holy tissue grail”) [67]. In gross examination, the cut surface of untreated EwS is grey-white, soft and frequently includes areas of hemorrhage and necrosis. Histologically, EwS has a solid pattern of growth, and is composed of monomorphic small cells with round nuclei [68]. The chromatin is finely stippled, and nucleoli are usually not apparent. In half of tumors, extensive deposits of glycogen are observed in the cytoplasm causing positivity in periodic acid–Schiff (PAS) staining. A ‘large cell’, or ‘atypical’, variant of EwS has been reported with larger-sized nuclei with irregular contours, conspicuous nucleoli, and usually PAS-negative stains (Figure 3) [69]. 

CD99 is a cell surface glycoprotein and a very sensitive but poorly specific diagnostic marker for EwS [70,71]. Strong, diffuse membranous expression of CD99 is evident by immunohistochemistry in ~95% of EwS [68]. However, CD99 expression occurs in many normal tissues and a wide variety of tumor types, including other SRCSs, and lymphoblastic lymphoma, and leukemia [71]. Hence, several, more specific or auxiliary immunohistochemical markers have been proposed. For example, the detection of FLI1 is relatively specific for EwS, but its specificity is limited by its expression in lymphoblastic leukemias and lymphomas, several soft-tissue sarcomas, and by the fact that around 15% of EwS exhibit variant translocations not involving *FLI1* [72]. Other markers such as Caveolin-1, NK2 homeobox 2 (Nkx-2.2), or combinations of immunohistochemical markers such as B-cell CLL/lymphoma 11B (BCL11B) and Golgi glycoprotein 1 (GLG1) have been proposed to support diagnosis of EwS, especially in cases negative for CD99 expression, but these require validation in prospective studies [71,73,74] (Figure 3).

Currently, the diagnosis of EwS can only be confirmed by molecular pathology being mandatory if cases have unusual clinical and pathological features [75]. FISH-based detection of *EWSR1* rearrangements and/or RT-PCR detection of *FET–ETS* gene fusions specific for EwS have been used for the past 25 years as a diagnostic tool [76]. Commercially available assays using *EWSR1* break-apart probes do not detect *EWSR1–ETS* fusions *per se*. Rather, these assays detect *EWSR1* rearrangements, which is important for differential diagnosis with other sarcoma subtypes that harbor *EWSR1* fusions with non-ETS genes (e.g., desmoplastic small round cell tumors (DSRCTs), or *EWSR1-NFATc2*-translocated sarcomas). In addition, FISH for an *EWSR1* break-apart can be misleading in malignant rhabdoid tumors (MRTs) in which commonly the genetic region encompassing *SMARCB1* is deleted that may involve the *EWSR1* gene, which is located close to *SMARCB1* on chr22 [77]. Hence, an immunohistochemical stain for INI1 (encoded by *SMARCB1*) should be considered, especially in very young patients or cases with congenital small-round cell tumors (SRCTs), and in which FISH may have indicated an *EWSR1* break-apart [77]. Loss of INI1 expression should then prompt further confirmation of the diagnosis of MRT. 

While rearrangements of *EWSR1* and *FUS* that are most commonly involved in classical EwS are reliably detectable in the majority of the cases by break-apart FISH assays, RNA-based approaches may be required in cases with particular gene fusions (e.g., *EWSR1-ERG*), which can be difficult to detect by routine FISH [78]. The same holds true for rearrangements with *CIC* and *BCOR*, where FISH and RNA-based analyses may be employed as complementary tools (please see section “Round cell sarcoma with non-ETS-fusions, and CIC/BCOR-rearranged sarcoma”). Nowadays, molecular genetic testing should be mandatory for diagnostic accuracy of sarcoma and appropriate clinical management, even when histological diagnosis is made by pathologist experts in this field [79]. Figure 3 illustrates the pathological workflow following radiological suspicion of EwS. The use of next-generation sequencing (NGS) is advisable for SRCSs in which FISH and/or RT-PCR cannot confirm the EwS diagnosis.

#### 2.3.2. Historical Evolution of EwS and EwS-Related Entities

In the last fifteen years, we have undergone a considerable revolution in the classification of round cell sarcomas. In particular, the introduction of NGS techniques has helped to define new entities that have been detaching from the general trunk of classical EwS [75]. Figure 4 graphically depicts this historical evolution. Historical concepts classified (peripheral) primitive neuroectodermal tumors ((p)PNET) and Askin tumors as entities apart from EwS, with the former showing a pronounced neuroectodermal phenotype with growth in rosettes and expression of at least two neuroendocrine markers (e.g., NSE, CD57), with Askin tumor being confined to the chest wall [80,81,82]. As it then became evident that these lesions shared *FET*-*ETS* gene fusions, diagnostic subgrouping was abandoned, and the 2013 WHO classification of sarcomas uniformly defined ‘Ewing sarcoma’ as an entity comprising the phenotypic spectrum of these tumors (Figure 4) [83,84].

#### 2.3.3. Round Cell Sarcoma with Non-ETS-Fusions and CIC/BCOR-Rearranged Sarcoma

The WHO classification includes the term ‘Ewing-like sarcoma’ (ELS). ELS—or, as we prefer to refer to these tumors, EwS-related entities—are a heterogeneous group of SRCSs being histologically similar to EwS. These ELS were considered as EwS until approximately 2010. ELS entities typically lack the hallmark EwS *FET–ETS* gene fusions but exhibit other recurrent and specific gene fusions/rearrangements. It is noteworthy that the term ‘Ewing-like’ was entirely based on morphological similarity with EwS, but recent RNA and methylation profiling approaches, as well as an increasing level of clinical evidence suggest that these rare, non-*FET* and/or non-*ETS* fusion-positive tumors are biologically distinct from *FET–ETS* EwS (Figure 4 and Figure 5) [71,84,87].

These SRCSs were previously considered as ‘histological variants’ of EwS and include *CIC*-fused and *BCOR*-rearranged sarcomas (Figure 5). *CIC*-fused sarcomas comprise sarcomas with *CIC–DUX4, CIC–FOXO4* and *CIC–NUTM1* fusions [88,89]. Many *CIC*-rearranged ELS express high levels of ETV4, which is a useful diagnostic marker in immunohistochemistry [74,88,90]. *BCOR*-rearranged sarcomas comprise sarcomas with *BCOR–CCNB3, BCOR–MAML3* and *ZC3H7B–BCOR* fusions, and sarcomas with *BCOR* internal duplications [84,91,92]. *NFATc2*-sarcomas include sarcomas with *EWSR1–NFATc2*, which commonly show an *EWSR1* amplification pattern on fluorescence in situ hybridization [93], and *FUS-NFATc2* fusions. The functional role of these gene fusions and rearrangements is currently being elucidated, and an active search for therapeutic targets is being carried out (for review please see [94]). Patients with ELSs may be eligible for EwS clinical trials because many of these ELS entities do not have specific clinical trials available (for review please see [94]). This reality may have important implications as recent data showed that patients with *BCOR*-rearranged sarcomas have comparable clinical outcomes to patients with EwS, whereas patients with *CIC*-fused sarcomas, which only rarely occur in bone, show relatively poorer outcomes and appear to be relatively resistant to chemotherapy, wherefore the application of neoadjuvant chemotherapy may need to be considered with caution in these patients (Figure 4 and Figure 5) [88,91].

Apart from the above-mentioned ELS, there is an increasing number of very rare ELS with translocations such as *EWSR1-SMARCA5*, *EWSR1-SP3*, and *EWSR1-PATZ1* [95,96,97], which do not cluster in unsupervised transcriptome analyses with EwS [84]. Although clinicopathological data on these tumors are scarce, analyses of recent case series suggest that *EWSR1-PATZ1* gene fusions may define a glioneuronal tumor entity [98], which appears to occur across a wide age range and which may show a predilection for the chest wall [95]. For all these ELSs, NGS approaches proved to be very useful to distinguish them from EwS and to identify the precise fusion genes (Figure 4 and Figure 5).

## 3. Local Therapy

### 3.1. Operative Local Therapy (by S. Collaud, J. Hardes, and A. Streitbürger)

#### 3.1.1. The Matter of Local Therapy—Scientifically Hard to Resolve, but Clinically Guided by Interdisciplinary Tumor Board Recommendations

Local therapy in patients with EwS is highly individualized. Patients should have the opportunity to explore local treatment options as soon after diagnosis as possible and decisions about local therapy should be made in collaboration with patients and families [17]. Expert interdisciplinary tumor boards are indispensable to define the optimal management in each individual case [99].

The optimal approach for local control in patients with EwS is influenced by a multitude of factors, e.g., patient age, tumor site, size, and local extension, and remains a matter of discussion. Randomized studies comparing surgery and radiotherapy (RT) in general, and their timing and sequence in particular, have either been limited or not been performed [100,101]. Still, in many studies, surgical resection seems to be superior to definitive RT for local control [100,102,103,104,105,106,107,108]. In consequence, a future randomized local control study does not seem feasible. For these reasons, many clinicians refer patients to both a surgical oncologist and a radiation oncologist to allow patients/families a full picture of risks and benefits of different local control approaches. Likewise, multidisciplinary tumor boards may play a role in medical decision making around local control in this disease.

The risk of local recurrence has to be carefully weighed against the functional outcome and late effects of either modality. For the latter, late effects such as growth impairment and second malignancies following RT have to be balanced against periprosthetic infections using tumor endoprostheses or pseudarthrosis after biological reconstructions, bone healing difficulties, and fractures following surgery [109,110,111].

#### 3.1.2. Surgical Strategies—Both Form and Function Follow Local Control

Definitive surgery typically follows an initial period of neoadjuvant chemotherapy, unless emergency surgical procedures are mandatory at diagnosis, e.g., in case of spinal cord compression. Importantly, patients should be referred to an experienced center for their operation. The timing of surgical local therapy is primarily dependent on the duration of neoadjuvant chemotherapy but is also determined by the availability of the necessary technical devices (e.g., custom-made implants, scheduling an irradiation appointment) and the most experienced interdisciplinary surgical and/or radiation therapist team [112,113]. Thus, it may seem better to maintain high-dose intensity by scheduling 1–2 additional courses of chemotherapy before adequate local treatment to achieve its best possible implementation logistically and technically.

The principles and techniques of surgical resection and reconstruction in primary malignant bone tumors have been defined by Enneking [114]. The aim of the tumor resection is a wide resection according to Enneking. An intralesional resection, e.g., debulking surgery, does not improve the prognosis, which emphasizes the necessity of comprehensive pretreatment imaging including whole compartment MRI (see above), as the surgical strategy is based on the initial tumor extension (please see section “Initial versus chemotherapy challenged tumor—operate to what extent?”).

Reconstructive surgical techniques should be applied wherever possible, but oncological control trumps limb preservation. The majority of patients with surgical local treatment requires bone reconstructive surgery. Biological reconstruction with bone grafts is one option, in particular in intercalary bone defects preserving the original joint. Techniques commonly used are vascularized fibula grafts, allografts or irradiated autografts. All of those excel at different advantages and disadvantages in terms of complication or reconstruction survival.

Today, tumor endoprostheses are widely used for bone reconstruction, in which a joint replacement is necessary. Endoprosthetic replacement in growing children requires specific expertise due to their rare indication: the small anatomy of the immature patients, the loss of the epiphysis in metaphyseal tumors require specific individualized implants. Custom made implants with small volumes adapt to the small anatomy. Non-invasive growing implants provide the opportunity to replace sacrificed growth plates and are able to compensate length discrepancies [115]. While reconstruction with modular tumor endoprostheses is the method of choice in adults, growth prostheses are used significantly less often due to the lower incidence of sarcomas in very young children and surgeons’ concerns including the soft-tissue coverage of the prosthesis and future revision operations [115]. While endoprostheses result in full stability of the involved extremity within short periods, good overall functional results, low rates of complications, and allow for rapid continuation of systemic treatment, they represent large foreign bodies and are vulnerable to complications such as deep infections which appear in up to 30%. Implant-associated infections are clinically relevant as they may disrupt adjuvant chemotherapy or result in amputation [116,117]. Noteworthy, post-operative RT has a negative impact on non-oncologic outcomes, especially infection and prosthetic failure, after endoprosthetic replacement of a long bone [17].

Primary amputation is rarely needed in EwS due to both tumor shrinkage during neoadjuvant chemotherapy and the availability of definitive RT as an acceptable alternative to surgery. Ablative surgery should be restricted to patients with severe complications after surgery and in tumors in which limb salvage would compromise the necessary surgical margin, e.g., extended infiltration of the neuro/vascular bundle, in cases of tumor progression under neoadjuvant treatment or in very young patients below the age of 3–6 years with tumors of the lower extremity. Secondary amputation is mainly due to local recurrence or periprosthetic infection. Comparative assessments of the quality of life in patients undergoing amputation or limb salvage surgery have not revealed any differences in long term outcomes [118]. Rotationplasty represents a special type of amputation of the lower leg, preferably for patients below the age of 6 years, offering the opportunity to maintain a functional hip and modified knee joint. This technique provides generally good long-term result in functional outcomes [119,120].

Regardless of the surgical method chosen, it should be emphasized that the long-term functional outcome of surgical local therapy depends on lifelong physical and psychosocial rehabilitation (e.g., physical therapy, occupational therapy, sports programs). Functional deficits do not occur only in the area of the reconstruction site. Often, adjacent joints, the contralateral extremity, or the spine (e.g., secondary scoliosis after hemipelvectomy) cause more problems than the surgical site [121].

#### 3.1.3. Surgical Margins and Histopathological Response to Systemic Treatment—Implications for Additional Local Therapy

The surgical margin status is a reliable indicator of tumor left in the patient [17]. An adequate surgical margin is one in which there is no viable tumor at the edge of the resection specimen that can be obtained by wide tumor resection, i.e., sufficient safety distance to the reactive zone of the tumor. Adequate margins significantly affect both the local recurrence rate and the overall survival [122,123]. Margins must be wide enough for optimal oncological control and narrow enough to maximize function. While the resection margin is not only based on quantitative information, e.g., mm, it also depends on the tissue quality of the boundary layer: If there is no infiltration, the periosteum, the perineurium, the vascular adventitia or fascia form a natural boundary layer, even if it is only a few mm wide; for example, in EwS of the distal femur, the adventitia may represent the anatomical margin to resection. In the same compartment, e.g., in the bone, a safety distance of 2–3 cm based on initial, pre-treatment radiological findings. In the AEWS1031 Ewing protocol, a clear margin is defined as no viable tumor at the cut surface.

In clinical practice, sufficient margins are carefully weighed against the functional outcome following surgery, in particular for pelvic EwS. For soft-tissue components, which usually shrink after neoadjuvant treatment, a downstaging strategy is discussed with the preoperative radiological detectable tumor guiding the extent of resection, while the same rules of intra-compartment margins apply. This strategy potentially omits areas of formal tumor seeding (please see section “Initial versus chemotherapy challenged tumor—operate to what extent?”). Noteworthy, these are non-evidenced based recommendations. So far, several studies have attempted to quantify the margins of resection, but no consensus has been reached, and debates are ongoing [17,124]. When analyzing the data in the literature, irrespective of sarcoma type, a threshold of >2 mm for a negative resection margin including an anatomical border structure, if possible, appears to be the optimal parameter for predicting local recurrence and can be chosen as an acceptable threshold to qualify surgical resection as safe (R0) [125].

An open biopsy channel must be completely included in the surgical specimen after open biopsy [56], while the risk of tumor seeding along the CNB tract may be lower but has yet to be fully determined (please see section “Biopsy”) [64,65]. 

Histopathological response has a major impact on local control rates in European studies [126]. While different criteria exist, an adequate response to chemotherapy should be taken as >90% necrosis [17]. Additional RT following surgery is recommended in any case of positive margins, while European protocols also recommend additional RT for narrow margins and/or poor histological response (≥10% viable tumor cells in the specimen) [67]. Similarly, combined local therapeutic strategies may be considered for large primary tumors with extensive soft-tissue extension [107]. Historically, RT can only be omitted when the area of original tumor extension has been included in the surgical specimen. Recent publications displayed that combined surgical resection and RT are associated with a higher overall survival probability in non-sacral tumors compared with surgery alone, even in patients with a wide resection and a good histologic response to neoadjuvant chemotherapy [127,128]; in contrast, different data questioned the need for additional RT in extremity EwS because of low percentages of local recurrence in extremity tumors and associated toxicity [129].

#### 3.1.4. Initial Versus Chemotherapy Responsive Tumor—Operate to What Extent?

Neoadjuvant chemotherapy alone or in combination with preoperative RT often results in a substantial reduction in tumor volume, especially of the extraosseous component, and facilitates or even enables adequate limb-sparing surgery. Persistent extraosseous tumor growth after neoadjuvant systemic treatment in patients with non-sacral EwS tumors undergoing surgical treatment might be an important indicator of reduced overall survival probability [127]. In patients with initially large tumors and very good histological response to chemotherapy, definitive surgery may exclude regions of original tumor growth unnoticed by the pathologist. By definition (please see section “Surgical margins and histopathological response to systemic treatment—implications for additional local therapy”), this would mean that adequate margins had not been obtained. Whether this has an impact on the rate of local recurrence and survival has not been elucidated so far, although a large initial tumor volume >200 mL has been repeatedly recognized to exhibit a negative impact on survival for EwS patients [9,101]. Several strategies are practiced: First, surgical resection based on initial imaging and growth/infiltration of the tumor in bone and soft tissue. Second, surgical resection based on preoperative, i.e., post-chemotherapy, imaging and growth/infiltration of the tumor in bone and soft tissue. Third, surgical resection of affected bone based on initial imaging and resection of affected soft tissues based on preoperative, i.e., post-chemotherapy, imaging. The latter strategy is based on the assumption that, on the one hand, initial imaging cannot always reliably distinguish displacing from infiltrative tumor growth and, on the other hand, the exact extent of soft-tissue infiltration after chemotherapy-induced tumor shrinkage may not be accurately recapitulated in situ. So far, prospective studies are missing to generate the scientific data needed for an evidence-based consensus strategy.

#### 3.1.5. Pathological Fracture in EwS

Pathological fractures are indicative of a large and highly aggressive tumor and the hematoma associated with the fracture could possibly result in spreading of the tumor into the surrounding soft tissues [130]. In EwS, fractures occur in 15% of patients with long-bone ES, most commonly in the proximal femur and at time of diagnosis [131,132]. If fractures occur after the completion of therapy, recurrence or second malignancy should be suspected [132]. In contrast to other sarcoma such as osteosarcoma, published data indicate that pathological fractures do not have a prognostic impact for EwS patients in terms of survival or local recurrence rate [131,133]. In cases of pathological fractures, wide resection is mandatory.

#### 3.1.6. EwS of the Extremities and the Role of Limb Perfusion

The standard of local care in patients with resectable tumors of the extremities has been considered definitive surgery which in most cases can be limb sparing [107]. Wide resection of the tumor within the involved bone in patients with non-sacral tumors may be associated with a decreased likelihood of local recurrence and improved overall survival [127]. Amputation in EwS is considered less often than for osteosarcoma. If resection of a distal leg tumor would lead to inadequate margins or a foot with poor function, amputation is indicated [17]. Other specific technical characteristics for the respective anatomical sites are summarized elsewhere [134].

Data on regional chemotherapy by perfusion of EwS of the extremities are lacking.

#### 3.1.7. Pelvic and Sacral EwS—When and How to Operate?

Historically, primary pelvic EwS had the least favorable prognosis compared with all other sites with higher rates of local relapse and reduced survival [135,136], but more recent studies published improved local control and overall survival rates for patients who underwent pelvic tumor resection or combined local treatment. EwS patients with large pelvic tumors (≥200 mL) appeared to benefit most of combined local modalities [137,138]. Currently, surgery for localized pelvic EwS is indicated if clear margins can be achieved, but in all other cases, the role of RT in single or combinatorial modalities for local control plays a greater role. Tumors which cross the midline in the sacrum or sacral tumors involving the S1 nerve route are not considered resectable because of the morbidity associated with surgery [17]. Current studies even refute the role of surgery for local therapy of pelvic EwS, favoring definitive RT for most sacral EwS [127,137,139]. Technical operability *per se* does not necessarily improve survival but may significantly limit the patient’s quality of life. Still, in the clinical care of pelvic and sacral EwS, the associated side effects of the local therapy modalities must be weighed against each other: While surgical intervention in small pelvic tumors, depending on their location, can mutilate and may lead to significantly reduced quality of life (e.g., impotence and incontinence), RT carries the risk of osteoarthritis. In this context, published outcomes for photon beam radiation may be surpassed by proton beam radiation in the future (please see section “Modern RT strategies and techniques”) [140] (Figure 6).

Even in patients with adequate margins and good histological response to the induction chemotherapy, some European guidelines suggest postoperative RT in pelvic EwS [141], though this approach has not been routinely adopted in North America. Other guidelines favor preoperative RT for these patients. In general, preoperative RT may be preferred when the tumor volume is large (Figure 6). As of now, recommendations for local treatment of pelvic EwS, in particular for disseminated patients with EwS primary of the pelvis, have to be made on a case-by-case basis.

If indicated, pelvic surgery in EwS has remained a challenge and should only be performed in experienced centers with case numbers above 10 patients/year [142]. Limb-saving techniques avoiding external hemipelvectomy is one of the goals in pelvic EwS surgery. Small pelvic endoprostheses are available for reconstruction. A re-stabilization of the pelvic ring is not necessary. Complications after pelvic sarcoma surgery can be reduced by avoiding skeletal reconstruction and choosing techniques like hip transposition as biological alternative after acetabulum resection [143]. To avoid a leg length discrepancy, hip joint transposition can be combined with endoprostheses [144]. In a recent study, acetabular reconstruction with hip transposition resulted in superior function in patients with pelvic EwS even when combined with preoperative RT, which improved tumor necrosis and rate of local control and survival [145]. Figure 6 provides a schematic workflow for local therapy decision in primary localized sacral/pelvic EwS.

#### 3.1.8. Primary Thoracic EwS 

Primary thoracic EwS includes EwS of the chest wall, lung or mediastinum. EwS represents the most common tumor of the chest wall in children and adolescents, which predominantly originates from bony structures of the chest wall (ribs, scapula, clavicle, sternum) and far less often from the soft tissue of the chest wall, the lung or the mediastinum [146]. The aim of surgery for thoracic EwS is complete tumor resection. It is best performed after induction chemotherapy to obtain tumor shrinkage and to facilitate complete resections, minimizing the need for additional large field RT, which is associated with a high risk of secondary morbidity including growth impairment, cardiomyopathy and secondary malignancies [146]. Resection of thoracic EwS should be performed in experienced thoracic surgery centers. At time of diagnosis, EwS of the chest wall may have a large soft-tissue component that fills the entire thoracic cavity; its possible bony origin is occasionally overlooked, which may lead to inaccurate treatment including surgical debulking procedures without neoadjuvant therapy [147]. Complete resection of the initially involved bony structures is recommended, while soft-tissue resection is guided by post-chemotherapy imaging and intraoperative findings. The extent of chest wall resection is a matter of debate. While some authors advocate complete removal of the involved rib together with their adjacent ribs [148,149], others advise resection of the involved rib only [150,151,152]. In a recently published retrospective study of 133 patients who underwent complete or partial resection of the involved rib, five-year event-free survival rates were similar for both groups [153].

In rare cases, EwS of the chest wall extends to the spine. Intraspinal infiltration is not a contraindication for surgery with curative intent *per se* if the tumor is deemed resectable. Tumor resectability needs careful preoperative multidisciplinary evaluation including orthopedic spine surgeons or neurosurgeons and thoracic surgeons. Noteworthy, the type of spinal reconstruction can affect the choice of RT treatment modality [17]. Figure 7 provides a schematic workflow for local therapy decision in primary EwS of the chest wall.

Chest wall reconstruction may be required to cover chest wall defects and preserve respiratory mechanics. Depending on the size and location of the chest wall defect, it is usually performed combining synthetic materials and/or pediculated/free musculo-(cutaneous) flaps. Primary soft-tissue closure is, however, possible in most cases [154]. While prevalence of long-term complications related to chest wall resection in pediatric patients is limited by small patient cohorts with relatively short follow up, scoliosis is a frequent complication in 25% to 43% of patients in this collective [155,156]. Risk factors for the development of scoliosis are patient age (either <6 years old or 12–15 years old) and resection of three or more ribs in the posterior sector. The type of reconstruction should prevent scoliosis and adapt to patient’s growth [155,156]. Satisfactory cosmesis is an important secondary goal.

Primary pulmonary and mediastinal EwS are very rare. They are described mostly as case reports or appear as single entity in larger studies [157,158]. In pulmonary EwS, principles of surgery are derived from lung cancer and primary pulmonary sarcoma surgery, recommending microscopically complete en-bloc tumor resection with surrounding involved structures if necessary. Anatomical lung resections (lobectomy, pneumonectomy) are favored over non-anatomical resections [157]. As for other primary pulmonary sarcoma entities, there is no evidence of a survival benefit to perform mediastinal lymph node dissection in EwS, although specific studies are missing. Nevertheless, as morbidity is very low and positive lymph node may have a strong impact on survival, it is recommended to improve pathologic staging [157].

Surgery for primary mediastinal EwS should lead to complete tumor resection with free resection margins. Due to close anatomical relation to unresectable anatomical structures (heart, esophagus, etc.) EwS of the mediastinum may not always be resectable [126,153,154,156].

#### 3.1.9. Patients May Benefit from Early Referral—Even Prior Diagnostic Biopsy—But Do Not Benefit from Re-Resection or Debulking Strategies

EwS is a rare disease and knowledge about diagnosis and treatment strategies determine patients’ survival and it is strongly recommended to refer patients with suspicion of a soft-tissue tumor to an experienced center. Perhaps more than for most cancers, the pathway to appropriate EwS treatment may already have been declared by events that have taken place prior to referral. For example, the type of biopsy that may have already been performed, or a prior inappropriate excision may jeopardize the form and outcome of local treatment thereafter [54,159]. Patients with EwS show a lower local recurrence probability if both the initial tumor biopsy and the tumor resection are performed at the same, preferably specialized institution [127]. Second-look procedures, neither for diagnostic purposes nor for attempting R0-resection, are not feasible for EwS due to sampling errors and missing benefit regarding patients’ survival, respectively. Although time to diagnosis does not appear to be associated with metastasis, surgical outcome, or survival, support in interdisciplinary decision making at specialty reference centers can affect overall patient’s survival [99,160].

In general, tumors which cannot be completely resected must be considered inoperable, since EwS patients do not benefit from debulking procedures [107]. Tumors may also be inoperable when they affect critical sites where complete excision would result in unacceptable mutilations or be associated with a very high risk of serious complications. Urgent surgery is recommended in patients with a possible EwS tumor of the spine which is causing neurological compromise. Without neurological signs, patients should have a biopsy before decompressive surgery to confirm the diagnosis [17].

#### 3.1.10. Disseminated and Relapsed EwS

Local therapy of involved sites is important for patients with primary, disseminated, multifocal EwS and should complement systemic treatment whenever possible. Outcome with local treatment of both primary tumor and extrapulmonary metastases is superior to results with local treatment of either the primary tumor or extrapulmonary metastases [8]. In disseminated EwS patients, surgery of both the primary tumor and extrapulmonary metastases yield to similar survival rates compared to definitive RT but combined-modality treatment is associated with a significantly better survival than single-modality local treatment [8]. Therefore, solitary bone metastases may be treated by surgery, RT or both if the morbidity is acceptable. Patients with pulmonary metastases should be considered for the same local treatment as those without [17]. 

Local control including surgery and/or RT may play an important role for some patients with relapsed EwS depending on the timing of relapse, location of relapse and sensitivity to chemotherapy. For localized relapsed EwS patients, local control strategies with either surgery or RT may improve outcomes and should strongly be considered when feasible [11,161], but are usually not feasible for those with widespread metastases. It is reasonable to consider aggressive surgery such as amputation or hemipelvectomy to treat locally recurrent disease if there are no metastases, even if prognosis may be limited [17]. For patients with a late isolated local recurrence, one study has demonstrated that patients who underwent aggressive surgery had significantly improved outcomes compared to those who did not. However, this was not a controlled study so the results must be interpreted with caution [162]. The role of pulmonary metastasectomy in relapsed EwS is controversial, with varying evidence of benefit between studies [163,164,165]. Metastasectomy is, however, commonly performed in selected patients with resectable pulmonary metastases and adequate cardiopulmonary function. The number of lung metastases, disease-free interval, and response to chemotherapy are well known factors influencing overall survival after pulmonary metastasectomy (in press: Stork et al., Number of metastases and their response to chemotherapy impact survival of patients with isolated lung metastases from bone-derived sarcoma. BMC Cancer). For patients with disseminated relapsed EwS, the role of surgery and radiation is largely palliative.

### 3.2. Radiotherapy (by B. Timmermann)

#### 3.2.1. Role of RT and Timing

The radiosensitivity of EwS has been recognized since its first description by James Ewing [166]. RT as an active modality for assuring local control is used as definitive RT in inoperable tumors or in combination with surgery, either pre- or postoperatively. Moreover, patients may benefit from RT in a palliative setting [167]. Still, RT for EwS requires specific interdisciplinary expertise. Results following interdisciplinary tumor board recommendations were superior to those without [99].

Reasons for preoperative RT include tumor progression or anticipated marginal or intralesional resection. The AEWS1031 Ewing protocol preferentially recommends preoperative RT for apparently resectable tumors in selected sites such as the pelvis, chest wall, and axially tumors where there is a higher risk of R+ resection. Postoperative RT is indicated in case of intralesional (including intraoperative spill) or marginal surgery (≥R1 resection). In European protocols, postoperative RT is also used in case of poor histological response (≥10% of viable tumor cells within resected tissue), regardless of surgical margins [105,107,168]. Nevertheless, the optimal timing of RT remains unclear for EwS, particularly when RT is combined with surgery. Both settings have advantages and disadvantages, e.g., impaired wound healing after surgery or suboptimal field sizes of RT. Randomized studies would be desired but are hampered by the need to evaluate response to chemotherapy.

#### 3.2.2. Modern RT Strategies and Techniques

In addition to achieving sufficient local tumor control, long-term tolerability and minimization of adverse effects of RT are major goals of modern RT techniques. Children are particularly vulnerable to radiation-induced late toxicities and secondary malignancies due to their immature tissue. Nowadays, smaller treatment area and consequently minimized dose burden to normal tissue are the results of individually optimized and very high precision RT application [169]. The optimal choice from various RT instruments may further reduce toxicity. Intensity-modulated RT (IMRT) and proton beam therapy (PBT) are typical modern instruments to offer optimal tumor coverage and sparing of critical structures [170]. In pelvic EwS, planning studies show superior dose conformity after modern IMRT as compared to conventional photon-based RT [171]. So far, results on modern PBT for EwS treatment are promising [172,173]. Early experiences suggest even higher conformity after intensity-modulated PBT (IMPT) when compared to IMRT, for example when looking at complex target volumes like chest wall lesions of EwS [174]. Axial tumors of craniofacial, spinal or pelvic sites may benefit considerably from PBT, due to the steep dose fall-off distally to the target as well as the relatively low number of treatment beams required for optimal dose conformity [175,176]. PBT has become a standard instrument of modern RT, particularly in young patients to reduce the risk of RT late effects and in cases of curative intent [162,177,178], but more data on the clinical evidence of PBT in comparison with conventional photon-based RT have to be provided by prospective comparative clinical trials [179].

With regard to the optimal fractionation regimen, hyperfractionated, accelerated regimens with, e.g., 1.2 Gray (Gy) per fraction twice daily, have been shown to have fewer long-term side effects and better functional outcomes for extremity or pelvic EwS [180,181]. However, their implementation in clinical practice is often limited by large radiation fields involving significant portions of the bowel, lung, and central nervous system and by concurrent high-dose chemotherapy [182].

#### 3.2.3. Primary Tumor Site, Prescription Dose, and Target Volume Definition

Combined strategies of surgery and RT yield to more favorable outcomes in local control than either definitive RT or surgery alone [104,128,183,184]. However, the optimal concept of RT within multimodal treatment is still under investigation. Modern regimes prefer definitive RT only for inoperable tumors that cannot be resected completely and in tumors of critical sites where complete surgery would result in unacceptable mutilation or is associated with a high risk of serious complications [5]. Previous reports indicated that RT as single modality results in a high incidence of local recurrence up to 35% [104,107,129], especially for large tumors. Meanwhile, international consensus has accumulated to treat most sacral EwS tumors with definitive RT as both the rate of local recurrence and the overall survival probability did not statistically differ between definitive RT and combined surgery and RT [127,129,137]. In case of intraspinal tumor extension causing neurological compromise, RT is usually indicated after decompressive surgery and should include the original tumor volume and all areas potentially contaminated by surgery [17]. In general, postoperative RT improves local control in case of incomplete removal of the tissues involved by the pre-chemotherapy tumor volume [128], but low percentages of local recurrence in extremity tumors and associated toxicity question the need for postoperative RT in extremity EwS [129].

Therapy for EwS is based on national and international study protocols (please see section “Systemic therapy”). Total doses between 45 and 60 Gy are typically prescribed for primary tumor sites with a 2 cm security margin [100]. Fractionation does not seem to affect local control [108]. Individual dose prescription will depend on the individual risk features such as extent of tumor but also on age and tumor site as critical organs at risk may prevent application of higher doses [185]. For example, in patients with a tumor extending into the intra-thoracic space or infiltrating the pleural cavity, the ipsilateral hemithorax including the ipsilateral lung is typically treated with lower dose radiation (≤20 Gy depending upon age and protocol) followed by a boost of radiation to the primary tumor area to complete the planned total dose (including the dose previously delivered to the hemithorax) (Figure 5). In this regard, either pleural involvement with a primary tumor or a pleural effusion in relation to a chest wall tumor may be an indication for postoperative RT [17]. In cases of initially large tumor extension with nodal as well as bilateral pleural or pulmonary involvement, an individualized RT concept should be tailored according to the specific situation on the advice of a dedicated interdisciplinary framework for EwS. Following this individual concept in RT, heterogeneity is inevitable in RT decision making which hampers clinical validation. 

Although the local control rate in the adjuvant setting has been around 85–90% in most series [107,127,153,186], there is no specific evaluation of different dose groups in relation to risk factors such as resection margin, tumor viability, size, and location. Moreover, there are some data supporting dose escalation with >60 Gy, especially for definitive RT [169]. Taking into account the unclear dose–response relationship in the adjuvant and definitive setting with different risk factors (initial tumor volume >/<8 cm, histologic response to chemotherapy, macroscopic residuals at the time of RT, and resection margin), the upcoming iEuroEwing (please see section “Intensified therapies—time but not dose or duration matters”) protocol attempts to establish a dose stratification that provides a higher boost dose of 9 Gy only for high-risk groups but offers moderate dose levels for low-risk groups.

Both initial and residual tumor area as well as areas of potential microscopic extent have to be covered by the target volume of RT. For tailoring target volume concepts, it is crucial to restrict higher dose levels to areas carrying the highest risk for failure [185]. Current RT concepts for EwS integrate so-called “cone down” techniques as well as “simultaneous integrated boost concepts” with restriction of higher dose levels to areas carrying the highest risk for local failure [187]. Traditionally, the scar from surgery is enclosed in the RT target volume in order to prevent treatment failure within the surgical access route. However, this may mean a significantly larger volume when compared to irradiating the tumor bed only. Nevertheless, as the impact of modern surgical procedures on risk of failure along the knife is not investigated so far, radiation oncologists are still reluctant to leave out scar regions, even if this may imply the risk of unnecessary “overtreatment” in some cases. With regard to the optimal RT volume concept, we unfortunately lack detailed data on pattern of relapse. Centralized plan review and the quality of applied RT has a positive impact on outcome [5]. However, quality assurance of plans, including retrospective or prospective plan evaluation, may be needed to analyze current dose-volume concepts more precisely in order to prevent over- or undertreatment in the future. More recently, courses on target volume delineation are offered by the ESTRO (European Society of Radiation Oncology), the PROS (Pediatric Society of Radiation Oncology) and on a national basis.

#### 3.2.4. RT for Relapse, Metastases and Whole-Lung Irradiation

For disseminated EwS, RT can be important component of multimodality therapy. In addition to the primary tumor, all extrapulmonary sites should be attempted to receive small volume irradiation of about 45 Gy. If this policy would exceed the volume of 30% of the patient’s bone marrow, local field RT should be restricted to the most relevant areas in order to avoid extensive bone marrow suppression. Autologous bone marrow transplantation may be considered after extended irradiation. Particularly for bone metastases in EwS, RT was effective to achieve local control [188]. With widespread bone metastases, RT to bony metastases is indicated when symptomatic. In case of oligometastatic disease, RT alone may be used in terms of stereotactic body RT [17]. Isolated cerebral metastases can be treated with 30 Gy (5 × 2 Gy/week) whole brain irradiation.

Whole-lung irradiation (WLI) is indicated if exclusive pulmonary metastases are present at the time of diagnosis, even when complete remission was obtained with chemotherapy [189]. Both lungs are to be irradiated to a dose of 15 or 18 Gy for patients of ≤14 or >14 years of age, respectively. The daily fraction dose is typically 1.5 Gy (Figure 5). In EwS patients with pulmonary relapse, WLI improves progression-free survival, particularly when good response to chemotherapy was achieved, i.e., pulmonary lesions resolve after systemic treatment [190].

In the relapsed setting, RT may be used similar to first-line strategies for EwS patients who relapsed ≥two years after the beginning of first-line therapy and/or who present with exclusive pulmonary metastases [11,191]. In a retrospective analysis of patients with isolated pulmonary relapse, the role of WLI was evaluated in patients who achieved a complete remission and who had not received WLI as a component of frontline therapy. While this approach did not improve overall survival, there was a trend towards improved progression free survival compared to patients who did not receive whole-lung radiation [190]. In light of some reported severe toxicities, the use of WLI should be evaluated on an individual basis according to the risk of pulmonary recurrence, respiratory comorbidities, and prior high-dose busulfan/melphalan (please see also “Radiation toxicity with high-dose treatment”) [190,192]. 

#### 3.2.5. Radiation Toxicity with High-Dose Treatment

Only a minority of patients showed significant pulmonary function abnormality after WLI [189]. The risk of adverse lung effects after WLI depends on several factors, including cumulative radiation dose and dose per fraction, high-dose chemotherapeutic regimen, and time interval between high-dose treatment and WLI [193]. The radiosensitization effect and toxicity of busulfan-containing chemotherapy before RT for EwS has been repeatedly described [192,194]. Severe toxicities leading to death have been observed in single patients who received high-dose large-volume RT following busulfan-containing high-dose treatment [193,194,195]. In patients with an indication for RT, the patient will not be offered busulfan-containing high-dose treatment if there are critical organs such as gut or lung in the fields, unless the technique can be provided which limits the dose to critical organs. If RT is mandatory, the time interval between stem cell reinfusion following high-dose chemotherapy and the start of RT should be at least 8–10 weeks (stable engraftment provided) to avoid rebound toxicity. Interestingly, no acute or chronic symptomatic pulmonary toxicities were observed in patients that received WLI after high-dose busulfan-melphalan, while grade 1 or 2 acute or chronic lung adverse effects were observed in up to 30% of patients that received WLI after high-dose treosulfan/melphalan or high-dose etoposide/melphalan regimens [193]. The safety of treosulfan in patients receiving radiation to the spinal cord or brain has not been established yet.

Historically, the potentiation of radiation pneumonitis by dactinomycin/actinomycin D has been well described both in vitro and in vivo [196,197,198,199], and current protocols omit actinomycin D during RT as this complication might still be of clinical significance when RT is applied concomitantly [200]. Similarly, various drugs included in current EwS chemotherapeutic regimen may potentiate radiation pneumonitis, although to a lesser extent, including cyclophosphamide and vincristine [201,202]. 

#### 3.2.6. Irradiation for Palliation

Local and disseminated recurrences in EwS typically have a dismal prognosis. Nevertheless, local treatment may provide medium-term local control, particularly in cases of preceding local failure. However, re-RT has to be evaluated carefully as previously applied doses and burden to important organs at risk have to be taken into account—at least with regard to short and medium toxicity. Currently, no standards for palliative RT in EwS are defined. Some studies on EwS include palliative RT in case of symptoms. One study revealed that RT is an effective treatment for metastatic EwS in palliative settings and offers a chance to provide symptom relief [203]. Pain, fracture or compression may trigger palliative RT, which can be given in a short course of hypofractionated RT [167]. Usually, small volume irradiation with 12 daily fractions of 3 Gy each is recommended.

## 4. Systemic Therapy

### 4.1. Evolution of the Current Systemic Backbone for Classical EwS (by U. Dirksen, S. G. DuBois, and D. S. Shulman)

#### 4.1.1. Development of VACA-Based Regimens—Multi-Agent Systemic Therapy Improves Outcomes

Prior to the 1970s, EwS tumors were treated solely with surgery and/or RT with nearly all patients eventually developing either primary or distant relapse [135,204]. Following the development of conventional chemotherapeutic agents in earlier decades, a number of small trials in the 1970s established a set of agents with activity against EwS. 

The first of these trials evaluated two combination therapies, vincristine and cyclophosphamide, and vincristine, actinomycin-D, cyclophosphamide and doxorubicin (adriamycin) (VACA), both of which demonstrated improved survival compared to historical controls [204]. Subsequently, a series of trials from the United States and Europe evaluated various VACA-based regimens as adjuvant, or neoadjuvant and adjuvant therapy, achieving five-year overall survivals that ranged from 49% to 79% [100,150,205,206]. In the Memorial Sloan Kettering experience, 18–20 months of vincristine, doxorubicin, and cyclophosphamide (VDC) achieved a five-year overall survival of 79% but utilized a cumulative doxorubicin dose of greater than 500 mg/m^2^ and carried an unacceptable degree of cardiac toxicity. Finally, the IESS-1 randomized trial in the 1970s compared VAC, VAC plus prophylactic whole-lung irradiation (WLI), and VACA, and demonstrated the superiority of the VACA arm over the other arms [207] (Table 1).

These early trials set the stage for a series of cooperative group trials that established the backbones of the current chemotherapy regimens used today throughout much of the world. Two important randomized cooperative group trials for patients with localized tumors, IESS-2 and CESS-81, established that shorter course of VACA with greater dose intensity in three-week cycles achieved equivalent or better outcomes to more protracted regimens with less dose intensity [103,208,209]. Multiple subsequent cooperative group trials in the US and Europe evaluated varying schedules of VACA at varying durations without significant improvements in outcomes. These trials as a whole provided an approximately 50% five-year overall survival for patients with localized EwS (Table 1). 

#### 4.1.2. Addition of Ifosfamide and Etoposide Further Improves Outcomes

Ifosfamide was established as an active agent against EwS in the 1980s. Three trials of patients with relapsed disease carried out by the United States National Cancer Institute (NCI) and the Pediatric Oncology Group demonstrated response rates ranging from 25% to 94% for ifosfamide or ifosfamide and etoposide (IE regimen) [210,211,212]. St. Jude Children’s Research Hospital added a three-cycle IE window to their neoadjuvant therapy and found that 95% of patients had an objective response to this therapy [213]. Memorial Sloan Kettering incorporated cycles of IE into their chemotherapy backbone that included VDC and found that EFS was 64% for patients with localized tumors, an improvement over historical control [214] (Table 1).

Subsequent cooperative group trials incorporated IE into VACA-based chemotherapy backbones [101,215,216,217,218]. The five-year OS in these studies improved to 60–70% for patients with non-metastatic disease. In North America, the INT-0091 randomized trial compared 3-week cycles of VDC to 3-week cycles of VDC alternating with IE [219]. This trial prescribed a cumulative dose of doxorubicin of 375 mg/m^2^, after which actinomycin-D was substituted. The five-year EFS for patients with non-metastatic disease in the experimental arm was 69% compared to 54% in the standard arm, establishing VDC/IE as the standard chemotherapy regimen in North America. In Europe, the EICESS-92 trial evaluated vincristine, dactinomycin, ifosfamide and doxorubicin (VAIA) neoadjuvant therapy with a randomization following local control in which standard risk patients were randomized to VACA vs. VAIA adjuvant therapy and high-risk patients were randomized to VAIA vs. VAIA plus etoposide [220]. There was no significant difference in either group. More recent European trials have utilized VIDE induction therapy, with VAI or VAC consolidation therapy [221] (Table 1).

#### 4.1.3. Intensified Therapies—Time but Not Dose or Duration Matters 

Subsequent cooperative group trials for patients with newly diagnosed EwS have attempted to intensify either the European or North American approach to systemic therapy. A number of prior trials had suggested that dose intensification improves survival, although gains in survival had come at the expense of increased toxicity. Memorial Sloan Kettering developed the P6 regimen, which utilized an augmented cyclophosphamide dose given in an alternating VDC/IE 7-cycle regimen. Forty-four patients treated with this regimen had a four-year EFS of 82% [222]. Subsequent approaches to intensification have intensified either the dose interval or utilized high-dose chemotherapy with stem cell rescue (Table 1).

North American cooperative group trial INT-0154 attempted to intensify doses of alkylating agents on a 30-week VDC/IE regimen versus a 42-week VDC/IE regimen, allowing for similar cumulative doses [105]. This approach did not improve outcomes and resulted in greater toxicity (Table 1).

The successor trial, AEWS0031, utilized interval compressed therapy, giving alternating VDC/IE cycles in 2-week intervals for 14 cycles [223]. The six-year EFS on the experimental arm was 73%, compared to 65% on the standard interval arm and toxicity was similar between arms. Based upon this result, interval compressed VDC/IE has now become the standard chemotherapy backbone across most centers in North America. The EURO-EWING-2012 trial compared VIDE induction to interval compressed VDC/IE induction [4]. Preliminary results demonstrated that the interval compressed VDC/IE had a high probability of being the superior regimen, with final analysis pending [6] (Table 1).

In Europe, following the French EW93 trial, which suggested a potential benefit of a consolidation strategy with high-dose busulfan and melphalan (BuMel) compared with conventional chemotherapy in localized EwS [224], both the Euro-E.W.I.N.G. 99 and Ewing 2008 international trials as collaboration of fourteen nations evaluated the role of high-dose chemotherapy with stem cell rescue for patients with high-risk localized tumors [7]. Patients received VIDE induction with post-local control randomization to receive consolidation with either cycles of VAI or high-dose BuMel followed by autologous stem cell rescue. High-risk localized patients were defined as patients with poor histologic response (≥10% viable tumor cells following induction), or tumor volume ≥200 mL in unresected tumors or initially resected tumors or patients with definitive RT for local control. In this trial, eight-year EFS was significantly superior in the BuMel arm at 60.7% vs. 47.1% in the standard therapy arm (Table 1).

Patients with isolated pulmonary metastatic disease were randomized to either six courses of VIDE and one cycle of VAI prior to either BuMel or seven courses of VAI and WLI. In this trial there was no clear benefit to BuMel over VAI and WLI [225]. The Euro-E.W.I.N.G. 99 trial investigated the role of high-dose BuMel for patients with newly diagnosed metastatic EwS. Patients with extra-pulmonary metastatic disease were non-randomly assigned to BuMel after VIDE induction. Outcomes were similar to those seen in prior trials in this population [9]. The Ewing 2008 trial randomized high-risk EwS patients with disseminated disease including extrapulmonary metastases to receive either VAC or VAC with high-dose treosulfan-melphalan (TreoMel) chemotherapy. No significant difference in three-year EFS was observed with 19.3% versus 21%, respectively [226]. Subgroup analyses showed that patients aged below 14 had a better outcome when treated with high-dose chemotherapy (three-year EFS 39% vs. 9%), which is supported by comparable results from the non-randomized Euro-E.W.I.N.G. 99 trial [9,226] (Table 1).

Through the EURO EWING Consortium (EEC), a harmonized protocol in Europe is targeted with iEuroEWING delivering a VDC/IE backbone [223], combined with a metronomic maintenance therapy as recommended by the European Pediatric Soft tissue sarcoma Study Group (EpSSG) for high-risk rhabdomyosarcoma [227], and a window for phase I/II trials.

#### 4.1.4. Adding Conventional Agents to Existing Backbone Regimens Has Not Thus Far Improved Outcomes

Successor trials have built upon these existing backbone regimens to add new chemotherapy combinations. Two North American trials have investigated the addition of topotecan and cyclophosphamide for newly diagnosed EwS. A phase 2 trial, COG 9457 evaluated an initial window of topotecan alone and in combination with cyclophosphamide [228]. While topotecan had little activity as monotherapy, six of 17 patients treated with the combination had a partial response. This trial provided rationale for COG trial AEWS1031, a phase 3 trial evaluating the addition of cycles of vincristine, topotecan and cyclophosphamide (VTC) to standard interval compressed VDC/IE [229]. The preliminary results of this trial were reported in 2019 and the addition of VTC did not show an EFS benefit in patients with newly diagnosed localized EwS (Table 1).

A number of trials have investigated the use of platinum-containing regimens in the treatment of patients with newly diagnosed EwS. The ISG/SSG IX protocol utilized alternating cycles of vincristine, doxorubicin, and ifosfamide, with cisplatin, doxorubicin, and ifosfamide, with a five-year EFS rate of 58%, inferior to existing contemporary regimens [230]. The Brazilian Ewing1 trial exhibited a five-year EFS rate of 67.9% in this evaluation of carboplatin, similar to that reported in other contemporary trials [231]. In the latter trial, patients with low-risk clinical features (resectable tumor, normal LDH) received consolidation therapy with VDC/IE, while all other patients received VDC/IE with two additional cycles of ICE (Table 1).

### 4.2. Systemic Treatment of Relapsed Classical EwS Including Combination Therapies (by S. G. DuBois and D. S. Shulman)

#### 4.2.1. Approach Relapse Therapy

Care of patients with relapsed EwS remains a significant challenge and must be individualized for each patient’s disease and goals of care. Outcomes for this group of patients remain poor, with less than 25% being long-term survivors [10,11,191,233,234]. Time to relapse is the most important identified prognostic factor [11,191,233,235]. Patients who relapse in the first two years from diagnosis have an OS of less than 10%, and those who relapse after two years from initial diagnosis have an OS closer to 30%. Another important prognostic factor at time of relapse is the site of relapse. Patients with combined local and distant relapse fair especially poorly compared to patients with an isolated local or metastatic relapse [12]. These factors should be taken into account when evaluating goals of care with a patient and a treatment plan to align with such goals.

Most patients with first recurrent disease are treated with conventional systemic chemotherapy. Patients who demonstrate response to therapy may undergo local control to sites of recurrence, with either curative or palliative intent (see also subsubsections “Disseminated and relapsed EwS” and “RT for relapse, metastases and whole-lung irradiation”). Multiple chemotherapy regimens have been evaluated for efficacy in the setting of relapse and are now considered effective options at first relapse. For patients with either a particularly low chance of cure at first recurrence (e.g., a patient with progression during frontline therapy), or patients with second relapse and beyond, additional lines of therapy can be considered for palliation. These patients are also often candidates for phase I or phase II clinical trials of novel agents or combination therapies.

#### 4.2.2. Systemic Therapies for Relapsed Disease—Time to Relapse Dictates Novel Agents Versus Re-Challenge with Frontline Drugs

Multiple systemic therapies have been evaluated for patients with relapsed disease, including regimens containing agents utilized as part of front-line therapy. Particularly for patients with late relapse, chemotherapy regimens may utilize a combination of agents that were included in front-line therapy as well as new agents with a reasonable chance of response [236,237]. For patients with earlier relapses, newer camptothecin-containing regimens are commonly used. An ongoing randomized European trial for patients with relapsed EwS, rEECur, is evaluating four chemotherapy regimens: high-dose ifosfamide; topotecan and cyclophosphamide; irinotecan and temozolomide; and gemcitabine and docetaxel [238,239] (Table 2).

Two camptothecin-based regimens are used in clinical practice today: topotecan and cyclophosphamide; and irinotecan and temozolomide, with or without vincristine. Topotecan was not found to be as efficacious as monotherapy in this context [240,241]. However, when paired with cyclophosphamide, induced a response in 30% of patients with relapsed EwS [242,243]. A significant proportion of these patients who responded to this combination had received prior alkylator therapy. An early trial of the combination of irinotecan and temozolomide demonstrated a response rate of 28% [244], while a subsequent single institution case series showed a response rate of 69% [245]. This combination has the added benefit of being available as an entirely oral regimen, which may be desirable for some patients with relapsed disease. The irinotecan and temozolomide regimen was reported to have an objective response rate of 20% in the rEECur trial and was determined to have a low likelihood of being superior to either topotecan and cyclophosphamide or high-dose ifosfamide which together had a pooled response rate of 23% [238] (Table 2).

The use of high-dose ifosfamide has demonstrated efficacy in patients with relapsed EwS, even following ifosfamide-containing front-line regimens. As noted previously, ifosfamide was identified to be an active agent against EwS in the 1980s [210,211,212]. In a study of patients with relapsed EwS who had received standard dose ifosfamide during front-line treatment, 34% of patients had a response to high-dose ifosfamide (15 g/m^2^/course) [236]. High-dose ifosfamide is one of two remaining regimens in the rEECur trial for patients with relapsed EwS [238]. An amendment to add carboplatin and etoposide as a new arm is currently internationally submitted (Table 2). 

The combination of gemcitabine and docetaxel has been evaluated in multiple trials, with response rates of 14% to lower doses of each agent [246], and 66% in a trial that utilized higher doses [247]. This combination was shown to have a low probability of being superior to the other regimens being evaluated in the rEECur trial (Table 2). Recently, rEECur introduced the combination of carboplatin and etoposide with pending results. Of note, in patients eligible for additional RT, carboplatin may serve as radio-sensitizer [248].

In patients who respond to conventional second-line chemotherapy, additional high-dose treatment may contribute to further reduce the risk of further events in patients [249]. Both BuMel and TreoMel show comparable outcomes in the relapse setting, but TreoMel might be more compatible with RT (please see section “Radiation toxicity with high-dose treatment”).

#### 4.2.3. Maintenance Therapy in EwS

Few trials have evaluated maintenance strategies in EwS. The Italian Sarcoma Group (ISG)/AIEOP EWS2 Study evaluated the addition of maintenance oral cyclophosphamide and celecoxib for patients with metastatic EwS. This study included 71 patients who enrolled between 2009 and 2019, who received induction therapy, radiotherapy and/or surgery, followed by consolidative high-dose busulfan/melphalan + autologous stem cell rescue, whole-lung irradiation (12–15 Gy), and maintenance therapy for 180 days. The three-year EFS was 79% for lung or single bone metastatic disease and 19% for patients with multicentric metastatic disease. Follow up is ongoing.

A maintenance strategy with zoledronic acid was evaluated in the Ewing 2008 trial. Patients with standard risk localized EwS did not benefit from maintenance treatment with zoledronic acid as add-on to the VIDE/VAI/VAC backbone. The three-year EFS was 84% in the zoledronic acid add-on group compared to 82% without zoledronic acid, while addition was associated with a higher rate of renal toxicity [261].

Finally, the Latin American Pediatric Oncology Group conducted a trial studying vinblastine and oral cyclophosphamide as maintenance therapy in EwS, with results pending at this time.

No studies thus far have demonstrated definitive benefit to maintenance therapy in EwS.

### 4.3. EwS-Targeted Therapy—Low-Hanging Fruit or Unfair Rumor? (by S. Bauer, U. Dirksen, S. G. DuBois, J. A. Toretsky, and D. S. Shulman)

#### 4.3.1. Targeted Agents Will Be Necessary to Overcome the Limitations of Conventional Chemotherapy and to Reduce the Burden of Late Effects in EwS

The therapeutic success of chemotherapy in EwS has been remarkable if you compare today’s survival with that of the 1970s. However, no new drugs have been successfully introduced to newly diagnosed patients for almost 40 years. Fortunately, OS for patients with localized disease is approximately 80% based on optimization of the currently used chemotherapy drugs, such as interval compression. However, we have pushed our current regimen to the maximum tolerable intensity, while still failing to cure many patients and leaving survivors with a significant burden of late effects. Exploiting mechanism-based vulnerabilities appears the only way to further improve outcome or reduce the burden of dose-intense, side-effect prone chemotherapy.

A mechanism-based approach would utilize tumor-driving biologic insights that could include activating proto-oncogenes, inactivation of tumor suppressors or use tumor-specific genomic abnormalities to target toxin-bound antibodies or direct an immune-response to the cancer cell. The genomic landscape analyses of EwS demonstrate a paucity of recurrent mutations [262,263]. In fact, there are no recurrent abnormalities that include kinase mutations or gene amplifications. There are small numbers of consistent abnormalities such as deletions of *STAG2* (17%) or mutations/deletions of *TP53* (10%). Despite this, a number of kinase inhibitors have been tested in EwS, but none of these have been successful in phase 2 evaluation, including inhibitors against aurora kinase A [264], c-kit [265], and insulin growth factor-1 receptor (IGF-1R) kinase (Table 2). 

#### 4.3.2. Adding Targeted Therapies to Existing Backbone Regimens

Few trials have evaluated targeted therapies for patients with newly diagnosed EwS. While no activating mutations are observed [266], the IGF-1 axis has historically been thought important in the pathogenesis of EwS and is frequently overexpressed [267]. In contrast to many preclinical studies on various pathways, the clinical testing of IGF-1R inhibitors translated into clinical benefit—unfortunately only in smaller number of patients. Multiple early phase trials of IGF-1R inhibitors have yielded consistent response rates of approximately 10% [252,254,255]. In the largest such trial to date, ganitumab, an IGF-1R monoclonal antibody inhibitor, was evaluated in combination with interval compressed VDC/IE in a randomized phase 3 trial by the Children’s Oncology Group (trial AEWS1221) [232]. The study included patients with metastatic classical EwS and did not demonstrate an improvement in EFS or OS with the addition of ganitumab. To date, strong predictors of response to IGF-1R inhibitor therapy have not been identified. Nonetheless, clinical activity as single therapy even without a strong biomarker is noteworthy and led the way to combinatorial approaches. A combination study of an IGF-1R inhibitor and an mTOR inhibitor yielded a response in 29% of patients [268]. There is now an ongoing study evaluating the combination of IGF-1R inhibition and CDK4/6 inhibition based on promising pre-clinical activity [269]. Targeting the insulin receptor A more specifically might present an alternative approach in the future [270].

#### 4.3.3. Established and Emerging Targeted Therapies for Relapsed Disease

At this time, effective targeted therapies for patients with EwS are in development, but none are currently part of standard therapy. While sequencing of tumors should continue to be pursued, tumor sequencing at relapse typically does not yield targetable genomic lesions aside from the canonical fusions that heretofore have not been targetable [262,263,271]. Despite these genetic realities, continued testing of new drugs has resulted in some potential therapeutic inroads.

Cabozantinib, a multi-tyrosine kinase inhibitor targeting primarily VEGFR2 and MET was recently evaluated in patients with relapsed EwS. MET signaling has been shown to be important in EwS tumorigenesis, and VEGF signaling for growth and metastatic potential [272,273]. A collaborative phase II trial between the US National Cancer Institute and the French Sarcoma Group known as CABONE evaluated cabozantinib in patients with relapsed EwS [251]. During the trial period, 39 evaluable patients with EwS were treated and 26% had an objective response with 33% alive at the median follow up of 31 months. Cabozantinib, where available, can be considered as a second-line agent for patients with relapsed disease, or at time of first relapse for patients with significant hematologic toxicity from front-line therapy or a desire for an entirely oral, non-chemotherapy regimen. Other multitargeted tyrosine kinase inhibitors (TKIs), including regorafenib, have shown activity in this context [250] (Table 2). Of note, following the early successes with kinase inhibitors, imatinib was tested in patients with EwS, who frequently show high KIT protein expression levels [274]. However, high expression levels do not predict activation of the kinase [275] which explains the lack of activity, even in a biomarker-restricted trial [276]. Further evaluation of TKIs in combination with chemotherapy is planned. 

Multiple studies have demonstrated that EwS tumors harbor deficiencies in DNA repair similar to BRCA 1/2 mutant cancers [277,278]. Poly(ADP-ribose) polymerase (PARP) inhibitors have demonstrated significant activity in BRCA-deficient tumors, as PARP inhibition leads to an accumulation of single-strand DNA breaks and eventually double-strand breaks, which these tumors cannot repair without homologous recombination. Further, pre-clinical work has demonstrated EwS cells are sensitive to PARP inhibition, especially in combination with temozolomide and irinotecan [279,280]. In a phase 2 trial of olaparib for adults with relapsed EwS, no patient had a response or durable stable disease [256]. In a Phase 1/2 study from the COG, talazoparib in combination with temozolomide was evaluated in patients with refractory solid tumors, including an expansion cohort for patients with EwS [257]. In this study, four of 15 patients with EwS had prolonged stable disease and no patients had an objective response. A phase 1 trial of talazoparib in combination with irinotecan with or without temozolomide was recently completed [258]. In this trial, 73% of the patients enrolled with EwS had a clinical response (1 complete response, 4 partial responses and 11 stable disease). Three quarters of these patients had previously been treated with irinotecan-based regimens and, given the lack of activity seen previously with talazoparib and temozolomide, the combination of talazoparib and irinotecan is thought to be an active combination. Further studies of PARP inhibitors will be needed to clarify these findings.

#### 4.3.4. The FET-ETS Translocations—A Clear Target, with Both Direct and Indirect Strategies

Clearly, the genomic marker that unifies EwS is a translocation that involves the *EWSR1* gene. EwS growth depends on the presence of the FET-ETS fusion [281,282,283]. The full roles of EWSR1-FLI1, and fluctuations in its expression level, in the growth and metastatic spread of EwS is an area of active research based on potential intratumoral heterogeneity [284,285]. EWSR1-FLI1 has a clear regulatory role in cell adhesion states, and this is modulated by potential matrix factors such as Wnt signaling [286]. These types of investigations require further study in patient tumors in order to validate the models.

Despite the attractive nature of targeting EWSR1-FLI1 fusion proteins, targeting the fusion is considered challenging as these proteins lack enzymatic activity and unlike kinases mostly lack obvious pockets for small molecules to bind [287]. Therefore, inhibition would require disruption of DNA-protein, protein-protein, or direct degradation of the fusion protein, e.g., using protacs. EWSR1-FLI1 activity depends on association with other proteins, where the interactions potentially exist over large surfaces requiring multiple contact points with few to no hydrophobic folds [288]. Thus, the biophysical interactions of EWSR1-FLI1 might include phase separation, leading to biomolecular condensate formation (a.k.a. ‘assemblages’) [289]. In a practical sense, inhibiting EWSR1-FLI1 is being addressed through approaches to both direct and indirect targeting that allow for small molecule targeting while progress in the biophysics continues [290].

TK216 is a first in class small molecule inhibitor designed to directly inhibit the interaction between the ETS-family fusion partner (i.e., FLI1) and RNA Helicase A (RHA), directly inhibiting the activity of the classical EwS fusion protein [291]. TK216 is an analog of YK-4-279 that was discovered empirically using surface plasmon resonance [292]. YK-4-279 is an enantiospecific inhibitor of the EWSR1-FLI1/RHA complex and this complex was validated as a therapeutic target [292,293]. TK216 is being evaluated in a phase 1 clinical trial in combination with vincristine (NCT02657005) based on preclinical synergistic findings with a TK216 analog [294]. A recent update of this trial reported two patients in long-term complete remission, supporting the preclinical EWSR1-FLI1 target validation [259].

Indirect targeting of EWSR1-FLI1 has been hampered by the absence of a detailed understanding of the mechanisms of action of the fusion. That has impaired development of indirect targeting of drugs to inhibit EWSR1-FLI1 and its cellular pathways [295]. Those compounds include chemicals that have chemotherapy-like structures (e.g., mithramycin analogs) [296], completely non-specific chemicals (e.g., arsenic trioxide or methylselenenic acid) [297,298]; drugs that broadly modify epigenetic regulation (deacetylase inhibitors such as romidepsin or depsipeptide or lysine-specific demethylase 1 (LSD1) inhibitors such as HCI-2509) [299,300]. So far, none of these approaches has resulted in a strong clinical benefit, presumably as the mechanism of action is less specific than preclinical studies suggested—resulting in a too small therapeutic window. A small molecule LSD1 inhibitor, seclidemstat (SP-2577) is under investigation in an ongoing phase 1/2 clinical trial in relapsed/refractory EwS (NCT03600649). Very recently, seclidemstat has completed the dose-escalation stage and established the recommended phase 2 dose.

Laboratory-based work has also elucidated novel functions of trabectedin to inhibit the EWSR1-FLI1 transcriptional program in a schedule-dependent manner [301]. A new clinical trial evaluating trabectedin in combination with irinotecan is currently underway (NCT04067115). In the laboratory, tremendous focus has been placed on new chemical degrader technology, with the hope that the fusion oncoprotein could be selectively degraded as a tool to treat this disease [302]. This line of investigation has not yet reached the clinic.

#### 4.3.5. Where Is Targeted Therapy Heading?

Effective clinical therapeutic targeting of EWSR1-FLI1 will require a deeper knowledge of its collaborating macromolecules (e.g., protein or RNA interactions). The mechanisms of how these macromolecules interact and modulate cellular programs that drive malignancy are at the forefront of research. Knowledge gained from continued cellular and animal models combined with training from clinical trial outcomes will be essential if patients with EwS are to have improved outcomes. The fact that targeting EWSR1-FLI1 is now in the clinic no longer relegates it to the ‘undruggable’ protein category.

#### 4.3.6. Challenges to Treat EwS in Low- and Middle-Income Countries

Many sarcoma patients in low- and middle-income countries are treated in hospitals lacking key infrastructure, including diagnostic capabilities, imaging modalities, treatment components, supportive care, and personnel. Abandonment and treatment-related mortality are additional challenges that complicate effective treatment and contribute to poorer outcomes of affected patients [303,304]. The standard therapies developed in parallel in Europe and North America are intensive and require extensive supportive care resources. Without the necessary resources to manage the toxicities, these intensive regimens may not be appropriate for use in low- and middle-income countries. Instead, modifications of these protocols may be needed on a country-by-country basis based upon agents available, cost to patients and the healthcare system, and available supportive care resources [305]. For example, metronomic chemotherapy combined with drug repositioning may represent therapeutic options for advanced, refractory, or relapsed EwS [306].

## 5. Scientific Perspectives on Clinical Enigmas of Disseminated EwS Disease

### 5.1. How Similar Are Primary Tumors with Metastatic Lesions? (by J. F. Amatruda and H. Kovar)

#### 5.1.1. Clonal Evolution of Metastases Seeds in Intratumor Heterogeneity and Correlates with Mutational Burden 

It has been known for long that presence of clinically overt metastases at either diagnosis or at relapse constitute the strongest adverse prognostic factor in EwS, and survival of patients with primary metastatic or relapsed disease is similarly bad. Yet, primary and secondary metastatic disease are not quite the same. Primary therapy-naïve metastases arise in untreated patients from disseminated tumor cells before diagnosis. Experience from the pre-chemotherapy era taught us that resection of the primary tumor-mass alone does not prohibit disease progression, and almost every EwS patient will develop metastases during the course of disease in absence of any systemic therapy. Thus, it is assumed that micrometastatic disease is generally present at diagnosis, but chemotherapy is able to eradicate disseminated tumor cells in the approximately 70% of patients lacking evidence of metastases at diagnosis [307]. This may imply that, at diagnosis, it is the high overall tumor burden, which makes the difference for patient survival in primary metastatic disease. In contrast, metastases occurring in relapsed disease of patients heavily treated with multimodal first-line therapy for initially localized disease likely developed from disseminated treatment-refractory tumor cell clones selected by and escaping chemotherapy. As a consequence, salvage therapy for relapsed disease is usually ineffective.

Even though EwS is most frequently diagnosed at adolescent or young adult age, much later than typical pediatric cancers, the frequency of somatic mutations is among the lowest in all cancers with on average only 6 to 11 somatic mutations [262,263,271]. This low mutation frequency may indicate tumor initiation at an early age, long before the clinical diagnosis. In fact, by looking at clock-like COSMIC mutational signatures 1 and 5 [308] in paired samples of primary tumors and metastases from the same patients, divergence of the metastatic clone from the primary tumor was found to have occurred at least one to two years before first clinical diagnosis [309]. It is thus not surprising that the mutational patterns of primary tumor and metastases/recurrencies may show only little overlap. Mutations predicted to be clonal or sub-clonal at diagnosis were not identified in the metastatic recurrent tumor samples. Instead, a portion of subclonal mutations at diagnosis became clonal at relapse, and recurrent metastatic tumors acquired new mutations [271]. DNA damage in response to reactive oxygen species [310] and uncorrected late replication errors during cancer progression [311] may be responsible for enrichment of COSMIC mutational signatures 18 and 8 in metastatic Ewing sarcoma. This finding and the observation of chemotherapy-induced additional mutational signatures in relapsed or secondary EwS (i.e., signature 31 after carboplatin treatment), suggest that new mutational processes drive relapse and metastasis in EwS [309]. In consequence, a significant increase in mutation rates from on average 0.37 in untreated primary tumors to 1.11 mutations per Mb in relapsed tumors was observed [271].

If the concept of early divergence and independent clonal evolution of metastases in metastatic EwS is correct, we might expect to see increased intra-tumor heterogeneity in the primary lesion of metastatic cases. This hypothesis received support from epigenome profiling of a large number of primary EwS. Here, intra-tumor DNA-methylation heterogeneity was higher in primary metastatic disease than in localized cases [312]. Additionally, on the genetic level, increased mutational burden, i.e., loss of the cohesion complex component STAG2 and *TP53* mutation, in bulk tumor analysis was found associated with inferior overall survival and time to progression [263]. In some tumors, *STAG2* loss was subclonal in the primary tumor, but characterized the main clone in the metastases. When viewed together with other studies, STAG2 mutation, present in about one fifth of EwS cases, may endow EwS cells with a more stem-like metastatic capacity [312,313].

#### 5.1.2. Lessons from Bulk Gene Expression Analyses Including Immune Contextures of EwS Tumors and in Peripheral Blood 

Since only about 20% of patients with primary localized disease relapse with distant metastases, it has been speculated for long that there may be a difference in gene expression of primary tumors that tend to relapse as compared to those that can be successfully cured. Until recently, however, it has been difficult to perform systematic studies on metastatic and relapsed disease, since standard-of-care for EwS does not typically involve re-biopsy of the cancer when the disease returns or has metastasized (please see section “Biopsy—the holy tissue grail”). Therefore, most comparative investigations of primary tumors with and without progression, and metastases were based on patient cohorts too small to allow for meaningful conclusions and used Affymetrix oligonucleotide arrays to analyze differences in gene expression patterns at low stringency. In each of these investigations, similar numbers of up- and downregulated genes were described, but with little overlap between individual studies. Results of these studies are briefly summarized in Table 3. 

The by far largest series of primary EwS analyzed for differential gene expression included patients from multi-centric American Children’s Oncology Group trials and European Intergroup Cooperative Ewing Sarcoma trials [318]. Supervised analysis of survivors vs. non-survivors in the COG study cohort revealed a small number of differentially expressed genes and several statistically significant gene signatures, which were strikingly restricted to tumors with stromal contamination (normal and reactive fibrovascular tissue and normal connective tissue into which the tumor cells had infiltrated) [318]. In line with tumor stroma interactions potentially playing a prognostic role, integrin pathway and chemokine signaling genes were found upregulated in stroma-rich poor-prognosis tumors, further supported by previous observation of CXCR4 and CXCR7 being upregulated in aggressive disease [322]. CXCR4 expression is known to be highly dynamic in EwS and upregulated in response to growth factor deprivation, hypoxia, and space constraints in the microenvironment [323]. Such tumor microenvironmental factors may also affect long-distance attraction of tumor cells to the lung, the most frequent site of EwS metastasis expressing high levels of the CXCR4 ligand CXCL12 [324].

### 5.2. How Immunogenic Are EwS Tumors and What Clinical Value Lies within? (by J. F. Amatruda and H. Kovar)

#### 5.2.1. Prognostic Immune Contextures of EwS Tumors and in Peripheral Blood 

In addition to stromal infiltrates, the immune contexture of EwS may affect disease progression. Regarding density, type, and distribution of infiltrating immune cells, considerable inter-tumor variation was reported. Results of these studies are briefly summarized in Table 3. An association of B-, CD8(+) T-, NK-, Th2-cell infiltrates or of neutrophils and M2 macrophages with poor prognosis was noted, whereas the infiltration with cytotoxic T-cells, M1 macrophages, mast cells, central and effector memory T-cells suggested that the patient had a good prognosis [319,320,321]. Our own studies identified increased protein expression of the metabolic sensor SIRT1 in the tumor cells of metastatic EwS [325]. Interestingly, on the RNA level, SIRT1 negatively correlated with gene expression of a so far uncharacterized immune infiltrate, consistent with a role for functional interaction between tumor and tumor-microenvironment compartments in EwS progression (unpublished).

Finally, gene expression profiling of peripheral blood cells in patients with EwS versus normal controls revealed that monocytosis and abnormal expression of CDH2 and CDT2 genes in the blood significantly correlated with poor patient prognosis, suggesting a systemic component in EwS disease progression, potentially through the intensification of osteoclastogenesis [326].

#### 5.2.2. Immunotargeting of EwS Tumors 

In recent decades, therapies that stimulate host immune responses to tumors have begun to produce significant clinical results, for hematologic malignancies and, increasingly, for solid tumors as well. Such therapies include immune checkpoint inhibitors (ICIs) such as anti-PD1/PD-L1 or anti-CTLA4 antibodies and cell-based therapies such as adoptive T-cell transfer and chimeric antigen receptor (CAR)-T cells [327]. Initial experience with ICIs for EwS has yielded few if any responses, though many trials are currently underway [328]. Immunotherapy for EwS must contend with certain challenges, some of which a recent review summarized as including lack of HLA class I expression in the tumor and an immunosuppressive tumor microenvironment due to the presence of myeloid-derived suppressor cells, F2 fibrocytes, and M2-like macrophages [329]. In addition to these antibody-based approaches, several cell-based immunotherapy strategies are currently investigated in clinical trials. Potential targets for T-cell based therapies include cell-surface molecules such as EGFR/HER2, IGF1R and ROR1 [330,331].

The ganglioside GD2, well-established as an immunotherapy target in neuroblastoma, may also have promise in EwS, when combined with other agents such as all-trans retinoic acid, EZH2 inhibitor or antibody targeting HGF [332,333,334]. The results of these and other preclinical studies and ongoing clinical trials are sure to provide new therapeutic options for Ewing sarcoma patients in the years ahead.

Altogether, so far bulk gene expression analyses did not reliably identify any EwS cell-intrinsic gene expression patterns discriminating between primary tumors of patients with localized disease, those presenting with metastases, and those at risk of treatment refractory metastatic relapse. However, these studies highlighted heterogeneity in the stromal and immune compartment affecting prognosis in EwS, and identified some candidate signaling pathways involved in potentially prognostic tumor microenvironment interactions.

### 5.3. Oncogene Plasticity—Myth or Tumor Strategy with Clinical Impact and Potential Therapeutic Consequence? (by J. F. Amatruda and H. Kovar)

#### EWSR1-FLI1 Oncogene Fluctuations as Metastatic Drivers in EwS 

Beyond inter-individual variation, the similarity of bulk gene expression patterns of primary tumors and corresponding metastases may not come as a surprise, as both disease stages are characterized by high proliferative activity. The major driver of tumor cell proliferation is the EWSR1-FLI1 oncogene present in most EwS cells of patients carrying the fusion. Experimental depletion of the fusion gene product results in loss of tumor cell proliferation but gain of migratory, invasive and metastatic properties as a result of epithelial-to-mesenchymal transition (EMT)-like cellular reprograming [285,335,336]. Among activated genes in response to EWSR1-FLI1 modulation are regulatory and structural cytoskeletal components including *ICAM1*, which was recently suggested as specifically associated with EwS invasion and metastasis [285,317]. In addition, EWS-FLI1^low^ cells are more chemoresistant, upregulate immune modulators PD-L1 and PD-L2, are resistant to T-cell mediated apoptosis, and upregulate angiogenic switch genes in response to Wnts [337,338]. We found MRTFB/YAP/TAZ/TEAD signaling as a master gene regulatory pathway specifically activated in EWSR1-FLI1^low^ cells [336], and treatment of EwS xenografts with the YAP/TAZ inhibitor verteporfin tended to reduce lung metastases in an orthotopic limb amputation xenograft mouse model [339]. In line, a recent study immunohistochemically identified an association of high YAP/TAZ expression with poor prognosis in a series of 55 tumors [340]. Using expression of EWSR1-FLI1-suppressed gene products as surrogate biomarkers, potentially metastatic EWSR1-FLI1^low^ tumor cells were estimated to exist in primary EwS at a frequency of about 1–2% [285]. Recently, single-cell RNA-sequencing of EwS PDXs confirmed fluctuations in EWSR1-FLI1 expression in a subpopulation of tumor cells associated with an EMT-like and hypoxic gene expression signature [284]. So far, it remains unknown what causes EWSR1-FLI1 fluctuations, if they are stochastic or occur in response to microenvironmental or intrinsic signaling cues. Future single-cell and spatial transcriptomic studies of human tumors might not only uncover potential prognosis-associated variations in frequency of EWSR1-FLI1^low^ cells in primary ES, but also elucidate potentially targetable interactions between distinct tumor compartments driving EWSR1-FLI1 fluctuations and EMT.

### 5.4. Why Does Outcome Differ between Lung Metastases and Metastases at Other Locations? (by J. F. Amatruda and H. Kovar)

#### 5.4.1. Therapeutic Accessibility

About one-quarter of EwS patients present with evidence of metastasis at diagnosis [341], and the presence of metastasis predicts worse clinical outcome. Current therapies for patients presenting with metastatic disease result in approximately 30% overall survival (OS) rates [9,342]. The most common sites of metastasis are lung, bone and bone marrow. Many trials have documented differences in response to treatment and overall survival in patients who develop metastases at different sites. In particular, patients whose sole metastastic site is lung have improved survival compared to patients with extrapulmonary metastatic disease. [343,344,345]. A Children’s Cancer Group and Pediatric Oncology Group study evaluating whether the addition of ifosfamide and etoposide to vincristine, doxorubicin, cyclophosphamide, and dactinomycin improved outcomes found that patients who had only lung metastases had an eight-year OS of 41%, respectively, compared to 30% OS rate for all patients [346]. Similarly, review of the European Intergroup Cooperative Ewing Sarcoma Studies showed that event-free survival (EFS) for isolated lung metastases was 34%, for bone/bone marrow metastases 28%, and for combined lung plus bone/bone marrow metastases 14% [343]. Other studies have also shown worse outcomes in patients with combined lung, bone and bone marrow metastases [9,347], and have identified presence of lymph node metastasis as an independent adverse prognostic factor [348]. 

What mechanisms explain the worse outcomes associated with metastasis at certain sites? In part this may be attributed to differential ability to apply therapy localized to the metastatic site. In the Euro-E.W.I.N.G. 99 study, patients receiving local treatment of both primary and metastatic sites had improved EFS compared to those who received either no local treatment, or local treatment of primary tumor only [8]. Several other studies also showed that irradiation of all metastatic sites improves EFS [188,349,350]. WLI has been effective for improving EFS in patients with lung as the sole site of metastasis [351]. As has been pointed out [352], these results must be interpreted cautiously, since patients with extensive bone marrow involvement or high burden of bony metastatic disease may have been ineligible for the types of surgery or RT that could be used to treat patients with lung-only metastases.

#### 5.4.2. Biological Concepts for Organotropism of EwS Metastasis

Beyond these clinical features, however, there is the strong possibility that different features of the local tumor microenvironment in different anatomic sites may modulate the response of metastatic cells to treatment, including systemic therapies. These metastatic niches differ significantly in their biophysical properties and cellular composition, which may differentially impact on the clinical behavior of metastatic disease, as the niche environment is well known to influence the invasive and metastatic potential and response to treatment of a tumor. The non-cellular solid component of tissue provides structural integrity and supports biochemical signaling, while its mechanical properties regulate how the host tissue contains or restrains the tumor [353]. The equilibrium modulus of the lung (0.5–3 kPa) is close to that of EwS tumor tissue (~2.3 kPa), while mineralized bone is several magnitudes stiffer (~15 MPa) [354,355]. Increasing tissue stiffness is a major component promoting transforming oncogene function [356], and one may therefore speculate that it also impacts on aggressiveness of bone metastatic disease. Hippo/YAP/TAZ signaling serves as the major cellular mechanosensor of the metastatic niche microenvironment, and increased YAP and TAZ protein expression and activity have recently been associated with low EWSR1-FLI1 levels and adverse disease outcome in EwS [336,339,340].

Another biophysical factor discriminating between lung and bone niches is oxygen tension. While the lung is a highly oxygenated soft tissue which may elicit mutagenic oxidative stress in tumor cells homing to it [357], bone and bone marrow provide a relatively hypoxic microenvironment despite being a highly vascularized tissue, but oxygen tensions fluctuate throughout the marrow cavity and across the endosteal and periosteal surfaces between 1% and 6% [358]. Hypoxia elicits signaling by hypoxia inducible factors HIF1α and HIF2α, whose activity affects a number of cellular processes, including glycolysis, angiogenesis, drug resistance, and many steps within the metastatic cascade [359]. In breast cancer, patients with disseminated tumor cells in the bone marrow tend to express higher levels of HIF1α in their primary tumors [360], suggesting that HIF1α signaling may promote tumor dissemination to the bone marrow. Among HIF1α targets is lysyl oxidase (LOX), which is also upregulated upon EWSR1-FLI1 fluctuations in EwS [285]. LOX has previously been demonstrated to remodel the pre-metastatic niche in the bone by enhancing osteoclastogenesis and inducing osteolysis in a spontaneous bone metastasis mouse model [361]. However, whether the hypoxic gene signature observed in EWSR1-FLI1^low^ EwS cells [284] contributes to increased metastatic propensity of these cells to the bone, remains to be demonstrated as the tail vein injection-model used to identify increased metastatic potential of tumor cells with transient EWSR1-FLI1 knockdown is not well suited to study bone metastasis [285]. However, it is able to monitor the propensity to colonize the lung. Lung tropism of metastasis may be a simple consequence of first pass of the tumor cells in the circulation and their entrapment in local capillaries of the lung. In this case, interaction between tissue-specific chemokines and tumor-associated chemokine receptors may provide circulating tumor cells an advantage to survive and grow in the new ligand-rich metastatic microenvironment. On the other hand, organs with high chemokine expression may guide tumor cells expressing the cognate receptor to that site for metastasis, and there is experimental evidence for chemokine signaling across endothelial layers resulting in directed tumor cell attraction to the site of metastasis. For example, the CXCL12 chemokine receptor CXCR4 is overexpressed by >80% of EwS. Both the lung and the bone marrow express high CXCL12 levels, and CXCL12 gradients may be responsible for long-distance attraction and binding to CXCR4-positive tumor cells. Other candidate chemo-attractants potentially involved in organ-specific metastasis include among others IL8 and IL6 (lung) and osteoblast-derived RANKL, OPN, BMPs, and IGF-1 and TGF-β (bone and bone marrow) [362].

On top of potentially chemokine-mediated tumor cell attraction by the metastatic niche, organotropism of metastases may be prepared by tumor-derived exosomes that carry a reprograming RNA and protein cargo to recipient cells at the site of metastasis. Lung specificity was reported to be mediated by specific α6β4 and α6β1 integrin patterns on the exosome surface, and exosome integrin uptake by lung resident cells resulted in activation of Src phosphorylation and pro-inflammatory *S100* gene expression promoting lung colonization of tumor cells [363]. Integrin or other exosome surface protein patterns specifically associated with bone/bone marrow metastasis have not been discovered so far. For EwS, secretion of exosomes and transfer of functionally active RNA and protein cargo to tumor and non-tumor cells has been reported [354,364], but studies investigating an association of specific exosomal patterns with differential organ tropism are not available.

Bone tropism of EwS metastasis is shared with common cancers such as of the breast, prostate, lung and kidney. Here, interaction with mesenchymal stromal cells in the metastatic niche may elicit a state of dormancy through BMPs and growth-arrest specific-6 (GAS6) protein secreted by the niche, permitting tumor cells to escape from anti-cancer drugs and immune therapy and emerge years to decades later as a clinically detectable bone metastasis [365,366]. In how far these specific mechanisms contribute to treatment-resistant bone/bone marrow metastatic disease in EwS, remains unknown. 

Upon homing to the bone/bone marrow, tumor cells disrupt physiological bone remodeling through the release of cytokines that stimulate osteoclastogenesis and resorption of the bone matrix and localized release of cytokines and growth factors (e.g., TGF-β), which further stimulate tumor cell proliferation [358]. Bisphosphonates targeting osteoclasts to block the functional dependency of tumor growth related to bone resorption have successfully been used to break this vicious cycle in preclinical EwS models [367,368], while the randomized phase 3 clinical trial Ewing 2008 showed no benefit of add-on zoledronic acid as maintenance therapy in terms of EFS rates in EwS patients [261].

Taken together, the mechanisms supporting colonization of the lung and the bone by metastatic EwS may differ significantly with consequences for the treatment. It remains unclear, if there is a tumor-intrinsic (epi-)genetic component determining organotropism of EwS metastasis, which may serve diagnostic staging. Orthotopic xenotransplantation models are required to address a potentially patient-specific tumor-inherent preference of primary EwS to metastasize to either the lung or bone, as conventional subcutaneous implantation models usually fail to metastasize at all [369]. Alternatively, innovative in vitro lung organoid models and scaffold models recapitulating the physical properties and cellular complexity of lung and bone niches may help to devise novel biologically targeted treatment options for patients with metastatic EwS. A bottleneck in all such studies is a shortness in untreated primary tumor material from diagnosis, which is usually obtained by fine-needle biopsy, and of viable material from the metastatic site after treatment (please see section “Biopsy—the holy tissue grail”). Establishment of subcutaneous PDXs may serve as a means to amplify this material largely retaining cellular complexity of the primary human tumor. However, early divergence and independent evolution of the metastatic cell clone, rare EWSR1-FLI1 fluctuation events driving EMT and tumor progression, and variations in stromal and immune cell infiltrates may have led to spatial heterogeneity in the primary tumor, which easily escapes detection in small biopsy materials. To account for spatial heterogeneity, and potentially allow for staging there is a need for obtaining biopsy material from multiple sites of the primary tumor. One way around this dilemma is the study of cell-free plasma DNA, which may provide a view on the mutational landscape of the full tumor burden in a patient, if deep sequencing is applied. In addition, as plasma DNA represents only the nucleosome-packaged fraction of the genome, it allows inference of open chromatin regions from depleted sequences as a surrogate of transcriptional activity [370]. Therefore, in the future, the study of cell-free plasma DNA may provide a valuable means to identify differences between lung metastatic and bone metastatic disease as a tool for early therapy staging.

### 5.5. Are We on Track with Preclinical Models? (by J. F. Amatruda and H. Kovar)

#### 5.5.1. Patient-Derived EwS Models

To improve outcomes for patients with metastatic EwS, preclinical models are critical, to support mechanistic studies of metastasis biology, and as a platform for testing new therapeutic approaches. Patient-derived cell lines have been a mainstay of EwS research, and a number of well-characterized cell lines are available and in wide use [371]. While important, there are widely recognized caveats in the use of long-established cell lines growing on rigid substrates, including genetic drift of cell lines under long-term culture [372] and non-physiologic response to 2D culture on plastic or glass. Several methods have been adopted to address these caveats, including analysis of primary patient-derived lines [373], and the development of 3D spheroid and organoid models [374,375]. Emphasizing the importance of modeling the tumor environment, Santoro and co-workers showed that culture of EwS cells on 3D scaffolds within a flow perfusion bioreactor promoted insulin-like growth factor-1 (IGF1) production and revealed shear stress-dependent resistance to an IGF-1R inhibitor [376].

There is widespread interest in the development and use of patient-derived xenografts (PDXs) as an alternative to long-established cells lines. In addition to more accurately representing the genetics of the primary tumor (at least during early passages) [377], PDXs may more faithfully represent the architecture of the primary tumor and the involvement of non-tumor stroma. PDX models have been established and used for individualized drug sensitivity testing [378,379,380,381,382,383]. To date, PDX models have not been extensively employed for the study of EwS metastasis, though it has been suggested that PDXs, especially when placed orthotopically rather than subcutaneously, may provide a more tractable model of metastasis for a variety of cancer types [384].

#### 5.5.2. Non-Patient Derived EwS Models

A different approach is needed to truly recapitulate the complex physiology of tumor cell interactions with the non-tumor microenvironment. A number of systems have been employed to allow in vivo preclinical modeling of EwS metastasis. One study focused on the contribution of focal adhesion kinase to EwS cell migration via its effects on focal adhesion formation and Rho-dependent cell migration. These investigators employed an avian chorioallantoic membrane model for in vivo testing of the effect of FAK1 inhibitors on EwS cell growth [385]. Another approach is the use of zebrafish embryos and larvae as a host for human EwS cell xenografts. The early-stage animals are optically transparent, making it possible to track metastatic cell dissemination with high resolution in a large number of test subjects. This method has been used to investigate a novel mechanism of NOTCH-induced tumor suppression involving modulation of the deacetylase SIRT1 [386]; to test the effects of the MDM2 inhibitor Nutlin-3 in combination with the small molecule YK-4-279 [387]; a role for receptor tyrosine kinase RON in EwS metastatic progression [388]; and to demonstrate synergistic effects of aurora kinase A inhibitors when combined with FAK inhibitors [389]. In one particularly notable study, Franzetti and co-workers demonstrated that plasticity in the expression level and activity of the EWSR1-FLI1 oncofusion affects the balance of proliferative states with those favoring migration, invasion and metastasis [285].

Mouse xenografts have been widely used for in vivo studies of metastasis. The most straightforward approach is the introduction of tumor cells into immunocompromised mice via tail vein injection, allowing seeding of metastatic sites, most commonly the lung. Using this technique, it was demonstrated that Caveolin-1 contributed to metastasis via regulation of matrix metalloproteinase production and activation and expression of SPARC [390]. Similarly, MMP-9 expression regulated by chondromodulin I contributed to the ability of EwS cells to seed the lung [391]. Another study found that silencing of EphA2 reduced tumorigenicity in vitro and incidence of lung metastasis [392]. A further example of this technique was the demonstration that overexpression of microRNA-138 led to downregulation of FAK1 and reduced lung seeding [393]. 

As a more natural model of metastasis, several studies have employed orthotopic xenografts. Tumor cell invasion and dissemination from a primary tumor established via intrafemoral injection was studied using multiple imaging modalities complemented by bioluminescence [394]. Using a similar approach, it was shown that zoledonic acid inhibited EwS cell invasion and metastasis, likely via down regulation of MMP-2 and -9 activities [395]. One variation of this technique involves implanting EwS cells in the hindlimb and allowing growth of the primary tumor and seeding of micrometastases. Removal of the tumor-bearing limb allows longer survival of the animal and the growth of spontaneous metastases. This approach was used to further demonstrate the importance of EphA2 [392] to show the anti-metastatic effect of treatment of the mice with the WNT pathway signaling inhibitor, WNT974 [396,397]. Another aspect of metastasis biology that has been explored is the role of immune cells, including the role of M2 macrophages promoting EwS [398]; the effect of expanded human NK cells in controlling primary and metastatic EwS [399].

#### 5.5.3. “The” EwS Mouse Model—Chronically Unavailable (So Far)

To fully mimic the in vivo environmental factors determining metastatic potential, including the 3D environment, stromal tissue mechanical factors, the role of vascular and lymphatic compartments and the host immune system, a genetic animal model of EwS would be highly valuable as a complement to patient-derived in vitro models and xenografts. However, multiple attempts to create genetically-engineered mouse models of EwS have not yielded a tractable model, likely due to the developmental toxicity of heterotopic expression of the oncofusion [400]. A zebrafish genetic model has been described, in which transposon-mediated expression of human EWSR1-FLI1 drives small round blue cell tumor formation with histologic and gene expression similarity to human EwS [401]. Further development of these and other genetically engineered models, particularly focusing on in vivo spread of tumor cells outside of the primary site, may add important mechanistic understanding of the metastatic process.

## 6. Biomarkers

### 6.1. Status Report on Biomarkers in EwS (by E. de Álava, M. Metzler, and V. Vieth)

#### 6.1.1. Diagnostic, Prognostic, and Therapeutic Markers in EwS

Proteins and metabolic products in the blood, which support tumor diagnosis and allow early detection of relapse such as AFP in hepatoblastoma and germ cell tumors or catecholamine metabolites in neuroblastoma, are not yet identified in EwS. Instead, direct tumor components that are released into the blood as a liquid biopsy in EwS like circulating tumor cells (CTCs), circulating tumor DNA (ctDNA), circulating tumor RNA, and extracellular vesicles (e.g., exosomes) containing tumor-specific biomaterials or signatures of the tumor microenvironment have been evaluated for their potential in clinical practice (Table 4) [402,403]. Prognostic biomarkers in EwS to predict the risk of progression or recurrence with associated outcome largely relate to histomorphological features and genetic markers including genomic, transcriptomic, and epigenetic alterations that are not implemented so far into clinical standards (e.g., STEAP1 expression, mir134a expression, genomic loss at certain genome sites). Currently, the status of disease represents the most robust clinical marker of prognosis, but within the collective of metastasized and/or recurrent patients, individuals with different response to therapy can currently neither be identified nor stratified. Studies are conflicting regarding the use of ^18^F-FDG-PET/CT as a prognostic tool in EwS [33]. Response to therapy is primarily assessed by imaging techniques following morphological criteria (tumor volume, morphological regression, i.e., necrosis). ^18^F-FDG-PET uptake shows a correlation between pre- and post-therapeutic standardized uptake value (SUV) and histological findings while providing data on additional quantitative parameters, such as metabolic tumor volume (MTV) and total lesion glycolysis (TLG) [404,405]. Still, efficacy of treatment after induction chemotherapy can only be objectively assessed by intratumoral histopathological evaluation (tumor viability, e.g., following Salzer-Kuntschik), which is absent if definitive RT is chosen. Therapeutic biomarkers indicating toxicity of both systemic and local treatment are still poorly uncovered.

#### 6.1.2. Limitations and Future Perspectives for EwS Biomarkers 

The prime and ongoing goal of biomarker research in EwS focus on identification and validation of biomarkers to stratify patients into specific clinical groups of EwS with similar prognostic outcome (e.g., depending on disease state), or which can help modulate therapy approaches (dose, intensity, novel agents) towards a more personalized medicine approach. Especially in the young(er) patient collective, better selection of risk groups is essential to avoid treatment-related morbidity for survivors.

A major disadvantage of EwS as a model disease to look for biomarkers is the extremely low incidence of this disease and the difficulties in conducting worldwide randomized trials in heterogeneous populations of such rare sarcomas. Several groups have conducted biomarker validations in recent few years (PROVABES, EEC-EuroEwing, COG) but these experiments have so far not converged into a common and consensus set of biomarkers (Table 4). Another major drawback is the apparently elusive nature of EWSR1-FLI1 as a therapeutic target (please see sections “The *EWSR1-ETS* translocations—A clear target, with both direct and indirect strategies” and “EWSR1-FLI1 oncogene fluctuations as metastatic drivers in EwS”). Nevertheless, the complete and constant presence of the underlying chromosomal translocation currently appears to be the most advanced marker even if its use as an individual molecular biomarker appears to be very resource intensive. However, alternative ctDNA-based methods using low-coverage whole-genome sequencing will emerge in the coming years and could be used in the clinic as a minimal residual disease proxy (Table 4).

## 7. Concluding Remarks (by Y. Uhlenbruch)

Despite significant efforts in both diagnostic and therapeutic strategies, many aspects of EwS remain elusive, emphasizing on the ongoing need for translational, international-coordinated and joined research. Currently, clinical decision-making balances between protocol-driven regimen, clinical facts, and interindividual concepts.

Recent decades have shown that tumor-specific strategies are difficult to implement. It is still non-directed chemotherapy controlling the disease for many patients but failing for far too many others. In the near future, a more personalized, translational approach embedded in prospective trial concepts is desirable and, even in the absence of targeted therapeutics, could mean personalize “treatment” without actually personalizing treatment.

## Figures and Tables

**Figure 1 jcm-10-01685-f001:**
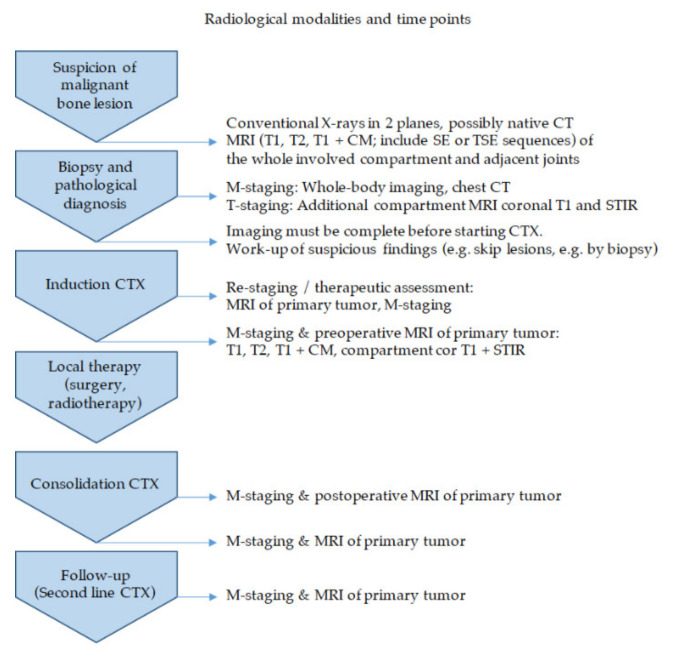
Radiological modalities and time points. The different stages of diagnosis and therapy are juxtaposed to the time points of radiological work-up with indicated modalities. CTX, chemotherapy; CT, computed tomography; MRI, magnetic resonance imaging; CM, contrast medium; SE, conventional spin echo; TSE, turbo spin echo; STIR, short tau inversion recovery sequence.

**Figure 2 jcm-10-01685-f002:**
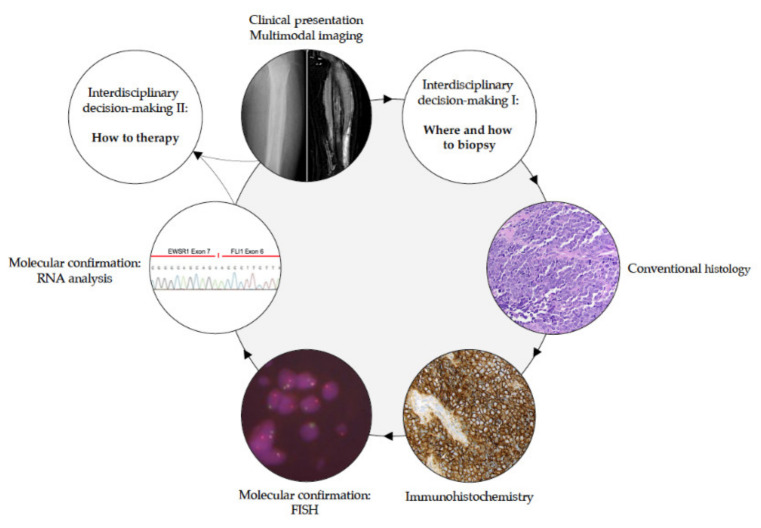
Schematic diagnostic workflow of EwS and related entities.

**Figure 3 jcm-10-01685-f003:**
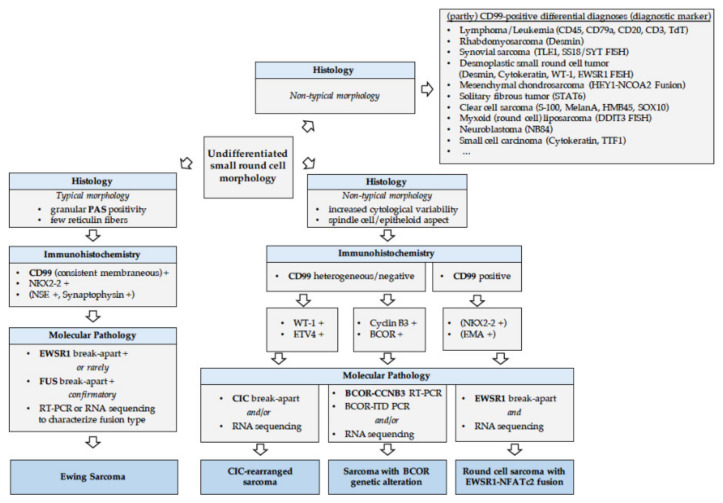
Schematic pathological workflow of EwS and related entities.

**Figure 4 jcm-10-01685-f004:**
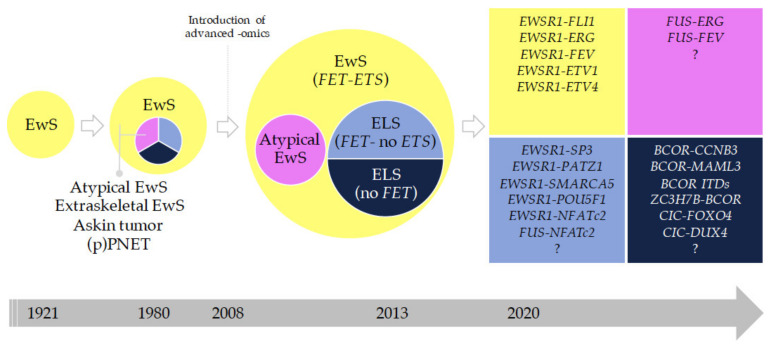
The broad spectrum of Ewing sarcoma (EwS) and related entities in historical evolution. Historical evolution of the concept of EwS and related entities. Several representative milestones are shown: 1980, introduction of the concept of atypical (large-cell) EwS and extraskeletal EwS [69]; 2008, introduction of mass sequencing [85]; 2013, fourth edition of the WHO classification of bone and soft-tissue tumors [86]; 2020, fifth edition of the WHO classification of bone and soft-tissue tumors [16]. The names in the circles indicate the genes most commonly involved in gene fusions of each sarcoma entity. ELS, Ewing-like sarcoma; ITD, internal tandem duplication; (p)PNET, (peripheral) primitive neuroectodermal tumors.

**Figure 5 jcm-10-01685-f005:**
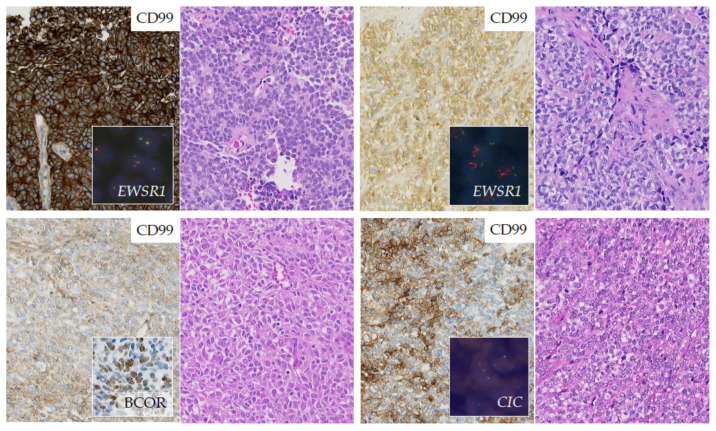
The broad spectrum of EwS and related entities in diagnostical work-up. Representative images of small round cell sarcomas showing classic EwS with uniform small round cells, strong expression of CD99, and aberrant patterns of *EWSR1* break-apart FISH, pointing to a genomic rearrangement (upper panel left). *EWSR1-NFATc2* translocated sarcoma with a more epithelioid cytology, positivity for CD99, and amplified isolated red signals in *EWSR1* break-apart FISH, suggesting rearrangement of the *EWSR1* gene associated with additional chromosomal aberrations (upper panel right). *BCOR-CCNB3* translocated sarcoma with primitive round to spindle cells arranged in sheets, heterogeneous CD99 expression, and positivity for BCOR by immunohistochemistry (lower panel left). *CIC*-rearranged sarcoma with a higher degree of cytological variability compared to classic EwS, inconsistent patchy CD99 positivity, and a classic break-apart in a *CIC* FISH assay (lower panel right).

**Figure 6 jcm-10-01685-f006:**
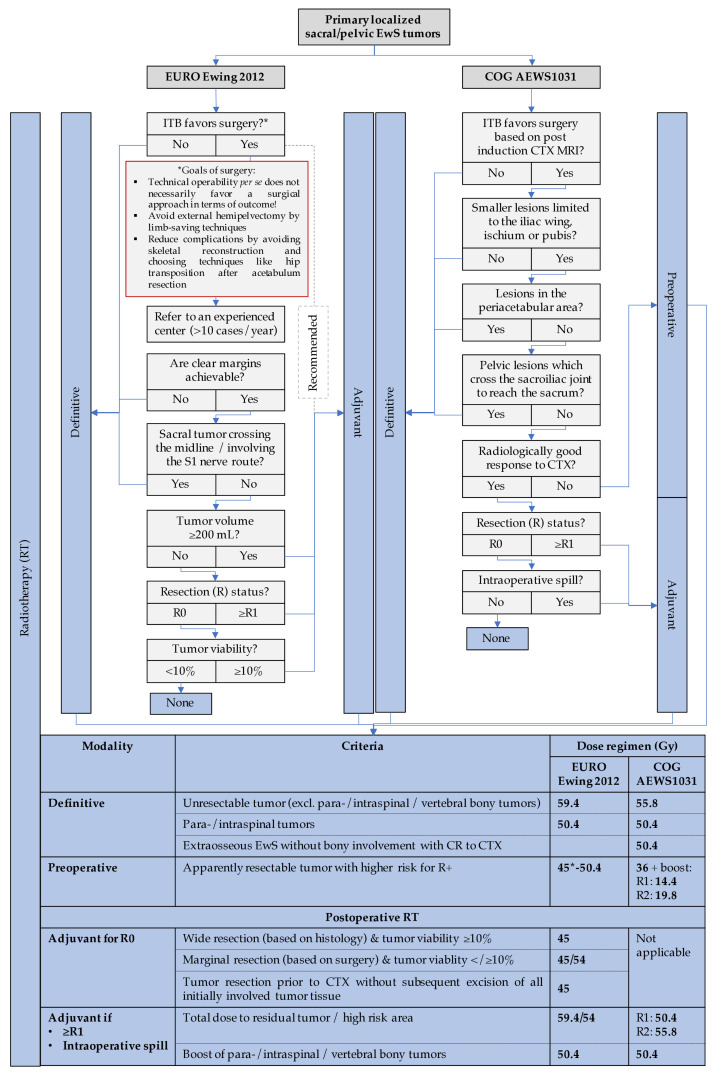
Schematic workflow for local therapy decision in primary localized sacral/pelvic Ewing sarcoma (EwS) and related entities based on EURO Ewing 2012 and COG AEWS1031 guidelines. The EURO Ewing 2012 recommends adjuvant radiotherapy for most sacral/pelvic tumors (dotted line). * 45 Gy if organs at risk (e.g., joints, myelon) are near the tumor. CTX, chemotherapy; CR, complete response; MRI, magnetic resonance imaging; R, resection status; RT, radiotherapy; TB, tumor board.

**Figure 7 jcm-10-01685-f007:**
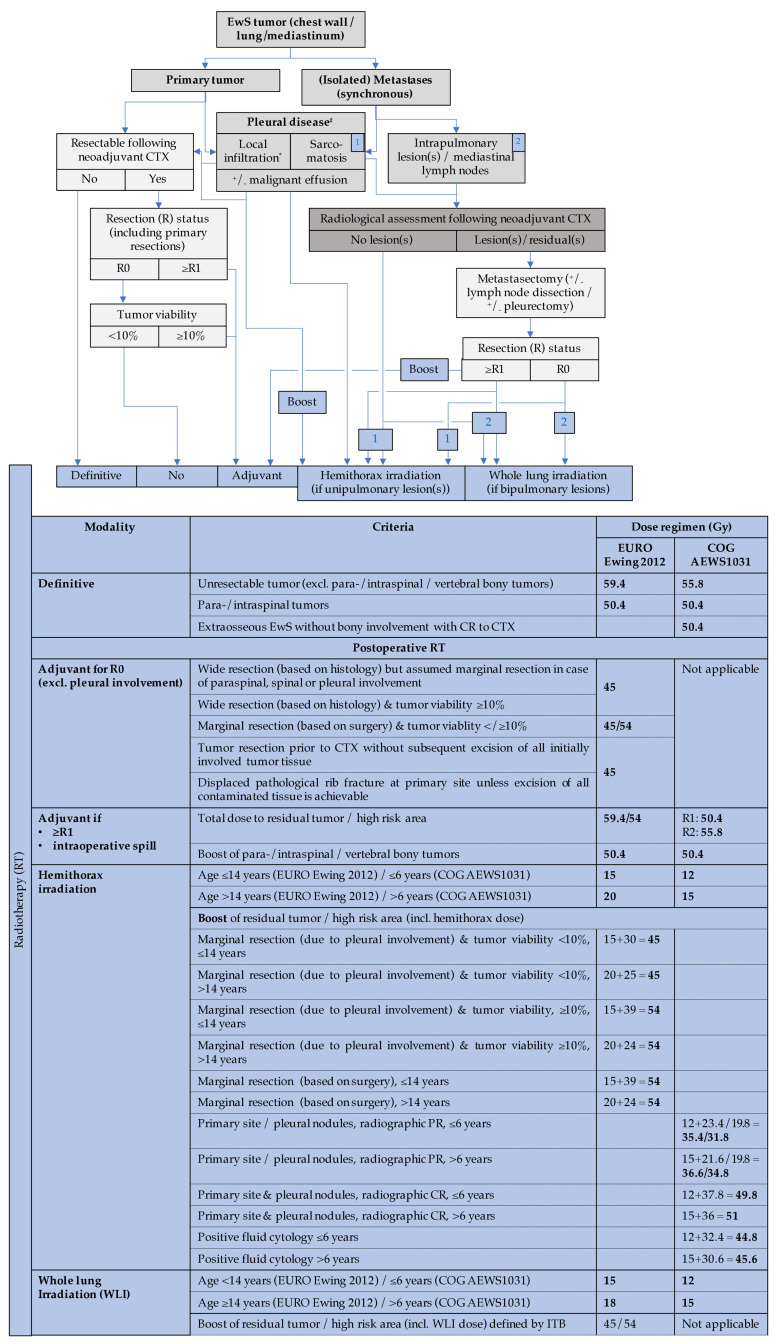
Schematic workflow for local therapy decision in thoracic/mediastinal Ewing sarcoma (EwS) and related entities based on EURO Ewing 2012 and COG AEWS1031 guidelines. Preoperative RT concepts are not included. * in case of local pleural infiltration, pleurectomy may be indicated. CTX, chemotherapy; CR, complete response; ITB, international tumor board; PR, partial response; R, resection status; RT, radiotherapy.

**Table 1 jcm-10-01685-t001:** Key trials establishing modern Ewing sarcoma (EwS) treatment.

Reference	Year of Publication	Chemotherapy Backbone Evaluated	Trial Name	Key Findings
[219]	2003	VDC vs. VDC/IE	INT-0091	Five-year EFS improved from 54% to 69% with the addition of IE to VDC for patients with **localized** EwS
[222]	2003	P6 (VDC/IE with augmented Cy)	MSK	Four-year EFS 82% for patients with **localized** EwS
[105]	2009	VDC/IE with augmented alkylator dosing vs. standard VDC/IE	INT-0154	No improvement in outcomes with alkylator dose intensification
[223]	2012	IC-VDC/IE	AEWS0031	Six-year EFS improved to 73% from 65% with interval compressed chemotherapy for **localized** EwS
[7]	2018	VAI vs. VAI/HD-BuMel	Euro-E.W.I.N.G. 99 and Ewing 2008	Eight-year EFS improved to 60.7% from 47.1% for **localized** high-risk EwS
[229]	2019	VDC/IE vs. VDC/IE/VTC	AEWS1031	No benefit to the addition of VCT cycles for **localized** EwS
[232]	2019	VDC/IE vs. VDC/IE/ganitumab	AEWS1221	No improvement in outcomes for **metastatic** EwS with the addition of ganitumab
[6]	2019	VIDE induction + VAI/VAC (or VIA/HD-BuMel) vs. VDC/IE induction + IE/VC (or VAI/HD-BuMel)	EURO Ewing 2012	VDC/IE induction was found on preliminary analysis to have superior PFS and OS compared to VIDE induction

VDC, vincristine/doxorubicin/cyclophosphamide; IE, ifosfamide/etoposide; IC-VDC/IE, interval compressed VDC; VAI, vincristine/actinomycin D/ifosfamide; HD-BuMel, high-dose busulfan/melphalan and stem cell rescue; VCT, vincristine/cyclophosphamide/topotecan; VIDE, vincristine/ifosfamide/doxorubicin/etoposide; VAC, vincristine/actinomycin D/cyclophosphamide; VC, vincristine/cyclophosphamide.

**Table 2 jcm-10-01685-t002:** Published studies of conventional and novel agents for patients with relapsed classical Ewing sarcoma (EwS).

Reference	Year of Publication	Agents Evaluated	Key Findings
Trials of conventional chemotherapy for relapse			
[236]	2009	High-dose ifosfamide	High-dose ifosfamide is active in relapsed EwS previously treated with standard-dose ifosfamide
[242]	2006	Topotecan/cyclophosphamide	33% of patients with PR and 27% with SD
[243]	2000	Topotecan/high-dose cyclophosphamide	Responses seen in patients with EwS
[244]	2004	Irinotecan/temozolomide	Responses seen in patients with EwS
[245]	2009	Irinotecan/temozolomide	63% ORR
[246]	2012	Gemcitabine/docetaxel	Limited activity seen in patients with EwS
[238]	2020	High-dose ifosfamide vs. topotecan/cyclophosphamide vs. irinotecan/temozolomide vs. gemcitabine/docetaxel	High-dose ifosfamide and topotecan/cyclophosphamide arms superior to irinotecan/temozolomide and gemcitabine/docetaxel. Enrollment is ongoing.
Trials of targeted therapies for relapse			
[250]	2017	Regorafenib (REGO)	10% objective response rate and a median PFS of 3.6 months
[251]	2020	Cabozantinib (CABONE)	26% ORR
[252]	2010	Figitumumab	2/16 patients with PR
[253]	2011	Figitumumab	14% of patients with PR
[254]	2011	R1507	10% ORR
[255]	2012	Ganitumab	6% ORR
[256]	2014	Olaparib	No patients with objective response
[257]	2020	Talazoparib/temozolomide	No patients with objective response, 15 with SD
[258]	2020	Talazoparib/irinotecan ^+^/_−_ temozolomide	73% of patients had a clinical response (1 CR; 4 PR; 11 SD)
[259]	2020	TK216	Two CR have been reported Enrollment is ongoing
[260]	2020	Seclidemstat	Enrollment is ongoing

SD, stable disease; ORR, objective response rate; PFS, progression-free survival; PR, partial response; CR, complete response.

**Table 3 jcm-10-01685-t003:** Summary of bulk gene expression analyses in Ewing sarcoma (EwS) tumors.

Patient Cohort (Number of Patients)	Genes/Pathways/Cell Infiltration in Patient Tumors with Poor Prognosis		Reference
	Enriched	Downregulated	
Localized tumors– non-progression (*n* = 7) vs. progression (*n* = 7)	▪ cell cycle ▪ invasion ▪ metastasis	▪ tumor suppressors ▪ inducers of ▪ apoptosis	[314]
Localized tumors – non-progression (*n* = 13) vs. progression (*n* = 17) additional 12 tumors for validation	▪ *MGST1*		[315]
Metastatic and localized tumors (*n* = 27)- non-regression (*n* = 7) vs. regression (*n* = 20) by chemotherapy		▪ Wnt ▪ angiogenesis ▪ apoptosis ▪ ubiquitin ▪ proteasome ▪ PI3 kinase ▪ p53	[316]
Primary tumors (*n* = 5) vs. unrelated metastasis samples (*n* = 6)	▪ *ICAM1*		[317]
Primary tumors (*n* = 56; additional *n* = 39 as validation cohort)– non-survivors vs. survivors	▪ cell motility ▪ cell migration ▪ cell adhesion ▪ glutathione metabolism ▪ integrin and ▪ chemokine receptor ▪ *CXCR**7* (cave: restricted to tumorswith stromal contamination)		[318]
Primary tumors (*n* = 88; additional *n* = 57 as validation cohort)– relapse vs. relapse-free survival	▪ B-cells ▪ CD8 T-cells ▪ NK-cells ▪ Th2-cells ▪ *VEGFA* ▪ *MMP* *9* ▪ *CXCL* *8* ▪ *EGF* ▪ *IGF* *1* ▪ *CXCR* *4* ▪ *TGFB* *1* ▪ *EGFR* ▪ *SPP* *1* ▪ *ICAM* *1*	▪ cytotoxic T-cells ▪ macrophages ▪ mast cells ▪ central and effector memory T-cells	[319]
Primary tumors (*n* = 197)– non-survivors vs. survivors	▪ neutrophils ▪ M2 macrophages ▪ HIF1α	▪ T-cells ▪ NK-cells	[320]
(Therapy naïve) Primary tumors (*n* = 27)– progression vs. non-progression	▪ CD8 T-cells ▪ (tumor infiltrating) ▪ CXCL9 ▪ CXCL10 ▪ CCL5		[321]

**Table 4 jcm-10-01685-t004:** Biomarkers in Ewing Sarcoma (EwS). Summary of the most relevant blood-based EwS biomarkers, encompassing diagnosis, prognosis and prediction of response to therapy, along with relevant references and levels of evidence. BM, bone marrow; CTX, chemotherapy; ddPCR, droplet digital polymerase chain reaction; Dx, diagnosis; EFS, event-free survival; IL, interleukin; PET, positron emission tomography; PB, peripheral blood; Pts, patients; WGS, whole-genome sequencing.

Biomarker Characteristics		Patient Characteristics				Study Details	
Type	Category	Biomaterial	Number	Disease Status	Detection Rate (% Positive)	Conclusion	Reference
EwS-specific biomarkers							
Circulating tumor cells	Diagnostic/ therapeutic	PB, BM	16		PB, BM at diagnosis (1 pts, 6 pts) BM during therapy (2 pts)	EwS cells in BM or PB can be identified by RT-PCR	[406]
	Diagnostic/ therapeutic	PB, BM	28	Primary and recurrent	PB, BM in non-metastatic pts (25) PB, BM in metastatic/ relapsed pts (50)	RT-PCR could serve as a EwS-specific marker of residual disease during CTX	[407]
	Diagnostic/ prognostic	Stem cell harvest; PB, BM	11		Stem cell harvest (100)	Number of cells may correlate with relapse after transplantation	[408]
Circulating tumor RNA (ctRNA)	Diagnostic/ prognostic	PB	28		68	Detection preceded progression; specific transcript types may affect progression	[409]
	Diagnostic/ prognostic	PB, BM	26		BM at diagnosis (43) PB, BM during follow up (58)	Occult tumor cells are strong predictors of recurrent disease in non-metastatic pts	[410]
	Diagnostic/ therapeutic	Tissue, PB	10		Tissue (83), PB (100)	EWSR1-FLI1 molecular diagnosis possible in PB even in absence of tissue; ctRNA correlated with ^18^F-FDG-PET parameters	[411]
Circulating tumor cells + circulating tumor RNA	Diagnostic/ therapeutic	PB, BM, PBSC	12	Metastatic	PBSC (2.5)	RT-PCR signal declines in PB and BM during CTX	[412]
	Diagnostic	PB	1				[413]
Circulating tumor DNA (by *EWSR1-FLI1* ctDNA PCR)	Diagnostic/ therapeutic		1			ddPCR to detect ctDNA could serve as a EwS-specific marker of recurrence	[397]
	Diagnostic/ therapeutic	PB	20	Localized and metastatic		Kinetics of *EWSR1-FLI1* ctDNA correlated with tumor volume	[414]
	Diagnostic/ therapeutic	PB	20		100	Combination of ^18^F-FDG-PET/CT and ctDNA quantification could serve as a EwS-specific marker for CTX response and relapse	[415]
Circulating tumor cells + ctDNA PCR	Diagnostic/ prognostic	PB, BM	Flow cytometry (109) PCR (225)		Flow cytometry (CD99(+), CD45(-)) (12.8) PCR (19.6)	Detection of micrometastatic disease by flow cytometry or RT-PCR is not associated with outcome	[416]
Circulating tumor DNA (by WGS)	Diagnostic	Tissue, PB	11		WGS (100)	ctDNA by both NGS and ddPCR could serve as a EwS-specific marker	[417]
	Diagnostic/ therapeutic/ prognostic	PB	11			ctDNA levels corresponded to CTX response	[418]
	Diagnostic/ prognostic	PB	94		ctDNA in localized pts (53)	ctDNA detection associated with inferior outcomes	[419]
	Diagnostic	Tissue, PB	2				[420]
Circulating cell-free mitochondrial DNA (ccf mtDNA)	Diagnostic/ prognostic	PB	25			ccf mtDNA levels associated with metastatic disease	[421]
MicroRNA– miR-125b	Diagnostic/ therapeutic	PB	63			miR-125b elevated in pts compared to healthy controls; miR-125b downregulation correlated with poor response to CTX	[422]
MicroRNA– miR34a	Diagnostic/ therapeutic/ prognostic	PB	31	Localized and metastatic		High miR34a inversely correlated with tumor volume; miR34a elevated in localized compared to metastatic pts; miR34a increased in localized pts at diagnosis and after end of CTX	[423]
Extracellular vesicles (EV)– EwS-specific transcripts	Diagnostic	Preclinical model				EVs could serve as a EwS-specific marker	[424]
Extracellular vesicles (EV)– EwS soluble EV proteome	Diagnostic/ prognostic	PB	10	Localized and metastatic			[425]
EwS-non-specific biomarkers (cytokines and other secreted peptides)							
IGF-1 and IGF-BP3	Prognostic	PB	22	Localized and metastatic		High baseline IGF-1 and IGF-BP3 associated with improved EFS; IGF-BP3 and IGF-2 increased during CTX	[426]
IL-6 and IL-8	Diagnostic/ therapeutic/ prognostic	PB	13	Localized			[427]
	Diagnostic/ prognostic	Tissue, PB	14			IL-6 elevated in some pts with poor prognosis	[428]
(Pro)cholecystokinin ((pro)CCK)	Diagnostic/ therapeutic/ prognostic	Tissue, PB	12	Primary and recurrent		ProCCK elevated in pts at primary Dx/ recurrence compared to pts during CTX; ProCCK correlated with tumor size	[429]
Pro-gastrin-releasing peptide (ProGRP) and neuron- specific enolase (NSE)	Diagnostic	Tissue, PB	9			ProGRP could serve as a EwS-specific marker	[430]
	Diagnostic/ prognostic	PB	16			ProGRP elevated in half of the pts; ProGRP reflected therapeutic response	[431]

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
