# Peer review of "Ewing Sarcoma—Diagnosis, Treatment, Clinical Challenges and Future Perspectives"

_jcm, 2021, doi:10.3390/jcm10081685_

Round 1

Reviewer 1 Report

 “Ewing sarcoma – diagnosis, treatment, clinical challenges and future perspectives” is a suitable title for the text and follows exactly the paragraphs it contains.

The text is very long and full of information so the introduction could be shortened and made more concise.

In paragraph 2.1 are provided information about the imaging evaluation of the tumor. Although the paragraph contains information about  the main imaging techniques to be used and their indication in the diagnosis and revaluation of the Ewing Sarcoma, there is not description of the aspect that the tumor may have on MRI evaluation.

In the numbering of the paragraph there is an error and on page 5 there is a second paragraph 2.1 which instead should be number 2.2. All subsequent paragraphs numbering must be changed after this correction. Also on page 5 and page 6, paragraphs 2.1.1., 2.1.2., 2.1.3. could be summarized in a single paragraph.

Paragraph 2.2 on Pathological diagnosis contains interesting information and is well developed. Figure 2 shows a lot of information but it is small and difficult to read.

Chapter 3 on local treatment is very long and could be summarized. Radiotherapy and surgery represented two fundamental times in the therapy of Ewing's sarcoma and therefore this chapter is of fundamental importance and could be more concise. Figure 5 on page 15 is really beautiful and useful for understanding the text. A similar image could also be constructed for paragraph 3.1.6. and for paragraph 3.1.7.

Chapter 4 carefully reports almost all the data relating to chemotherapy published to date.

Paragraphs 5.4.1 and 5.4.2 can be summarized in a single paragraph.

Chapter 6 does not add many new elements to the topics covered in the other chapters. In particular, chapter 6 could be reduced to just paragraph 6.1.3 and become a paragraph of chapter 5.

Chapter 7 may be out of context with the rest of the article.

The page numbering has an error and after page 38/62 the numbering starts again.

Reviewer 2 Report

This is a very comprehensive and impressive review written by “well-known” and merited people representing various disciplines in MD-teams mandatory to ensure the best current management of patients with Ewing sarcoma (ES). I congratulate the authors with this tremendous work. To peer-review such a voluminous paper is, indeed, a challenge (43 pages text and 412 references cited). The title is, indeed, very interesting and warranted; albeit ambitious. I hope my feedback may further improve this important review. To quote Benjamin Disraeli: “It is much easier to be critical than to be correct”.

Some of the points mentioned below are not meant as serious criticism, but rather suggested adjustments to better avoid the risk of blurring the practical/clinical message to come through. That said, several of the sections from Chapter 5 and onwards (e.g. preclinical models) are very long. This is rendering the reader in the challenging situation not to “drown in details”. The main justification of such a review is: What is, from a clinical management point of view, the currently best evidence-based standard of care and what is really new in this particular review? I note that authors cite very few of the other more recent ES-reviews and treatment guidelines. As an example, they do not give credit to the most recent one in bone sarcomas:  Casali et al., Ann Oncol. 2018 Oct 1;29(Suppl 4):iv79-iv95. (but rather one from 2014 (Ref. 64).

“Self-citations” is un-avoidable in the current review taking the string of authors into consideration, but I think it is “too much” in several of the chapters. This is leaving out contributions from other ES-groups – e.g. the French, UK experiences and also some published results from ISG/SSG IV study.

Of particular interest for the readers around the world (given the title “…..future perspectives”) is for the authors to provide a short description the two planned consortium trials “i EuroEwing” vrs. “InterEwing 1”. These prospective clinical studies will govern the “gold standard(s)” of EW multimodal treatment of ES in more than a decade to come. What are really the differences? The pressing questions aimed to be addressed and their respective justifications? How high radiotherapy dose is really justified (in the various scenarios/risk-adapted based of tumorbiological and clinical characteristics)? Isolated local recurrences (without metastatic relapse) are at present expected to be very low in a radio-sensitive cancer entity like ES (in combo with “state of the art” chemotherapy).

Since I think it is fair to say (also from reading this paper) that consensus largely exist regarding the combos of active drugs, their doses (and also the time-line/scheduling of chemotherapy-courses), I would like to have seen an even more critical discussion of individual case-tailored choice of “local therapy modality” (even if good attempts are already made from section 3.1.1 and throughout chapter 3). This is particularly related to axial locations (including pelvic/sacral, spine & H&Neck, skull-base ES).  Here the term “operable” is, in my own clinical experience, too often judged and governed as “technically operable”. This may relatively often lead to severe post-operative complications/infections that often is delaying the subsequent and mandatory need for adequate (high-?!) dose-intensity of chemotherapy. Furthermore, this may preclude the justified post-operative radiotherapy to be administered; timely, or even not to be given at all (based on status rel. to margins & histological response to chemo). As a result, this may significantly result in a dismal survival for such individuals. Then comes the very long and for the patients (and health care system) challenging physical and psycho-social rehabilitation phase.  With the current improvements in radiotherapy (including proton-therapy), I think there is an under-use of that modality as definitive local treatment. Relatively mutilating surgery is often “justified” based on a (not well studied – based on older studies/techniques) risk of radiation –induced second cancers (in the range of 5-7%, but in a 10-30 years later time frame – except from the very few Li-Fraumeni-cases). I am fully aware of the lack of “high level” evidence in the literature, but “eminence/experience-based” decisions are often necessary to deliver the best and individualized treatment for a given ES patient.

Also the “timing” of local therapy for primary tumor needs to be better discussed. It seems better to maintain high dose-intensity (scheduling of 1-2 extra courses of chemotherapy) waiting for the best /correct local treatment to be logistically in place (choice of modality and “best team”; either surgically or radiotherapy-wise).

Radiotherapy: I would like to see somewhat more related to benefit of proton therapy, impact fraction-dose and fractionation regimen (is 1.5Gy hyperfractionated/accelerated twice daily “dead”?).  Also a better justification for total radiation dose is welcomed and to provide a more dynamic approach to geometrical/anatomical margins.  Also the very effective stereotactic irradiation of oligometastatic sites should be mentioned. Related to indication for “total lung irradiation”, also a small but real risk of a fatal outcome has been published in osteosarcoma and ES patients and may deserve to be mentioned.

 Re. “maintenance therapy” 4.2.3: Is trofosfamide (Ixoten) as oral chrono-therapy regarded as a justified regimen in poor prognosis ES?

Details: Chapter 2.1 Imaging: PET is described as just “one modality” (e.g. lines 151, 152). Make sure to explain the difference between 18F-bonePET and 18F-FDG-PET and their respective roles. Page 5: Biopsy is listed as chapter 2.1 – should be 2.2.. adjusted accordingly – also in rest of the paper. Figure 1 is “blurred” with the “sign of radioactivity/ionizing radiation” displayed on top. This is misleading since MRI is among the most important radiological modalities. Figure 2 is not possible to read. The font/writing style of References need to be consistent – se ref 25, 218, 219, 225 as examples.

Lastly, would it be possible to voice a message for clinics around the “developing world”. What are the “silver and bronze standards”in ES?  Most ES patients around the world still do not have access to cutting edge multimodal treatment (several recent publications are available), or participation in prospective consortium clinical studies (where no financial incentives exist/will be in place for most of the recruiting hospitals).

A paradox (more “philosophical” comment not directly related to this well written paper): 

By first glance, this review looks more like a “text-book chapter” but I see that the authors have attempted to update and structure the document compatible with and acceptable for a clinical/scientific (assumingly invited?) review-paper. In the current “digital world” there is a rapidly growing plethora of “open access” journal (the actual one already having achieved a decent impact factor). That, at least, gives authors some academic credit, rather than just honors related to a textbook. What I dislike, however, is the “smart move” by the publisher that this impact factor is to some degree driven by “self-citations” and comes with a significant economic prize; paid by the authors as publication costs/open access fees (and…lastly, unpaid “peer-reviewers”!).
